# Tissue distribution and retention drives efficacy of rapidly clearing VHL-based PROTACs
Donglu Zhang [1] ✉, Bin Ma [1], Peter S. Dragovich[1], Li Ma[1], Shu Chen[1], Eugene C. Chen[1], Xiaofen Ye[1], Joyce Liu[1], Jennifer Pizzano[2], Elizabeth Bortolon[2], Emily Chan[1], Xing Zhang[1], Yi-Chen Chen[1], Elizabeth S. Levy[1], Robert L. Yauch [1], S. Cyrus Khojasteh[1] & Cornelis E. C. A. Hop [1] ✉

## Abstract

**Background** Proteolysis-targeting chimeras (PROTACs) are being developed for therapeutic use. However, they have poor pharmacokinetic profiles and their tissue distribution kinetics are not known.

**Methods** A typical von Hippel-Lindau tumor suppressor (VHL)—PROTAC [14]C-A947 (BRM degrader)—was synthesized and its tissue distribution kinetics was studied by quantitative whole-body autoradiography (QWBA) and tissue excision in rats following IV dosing. Bile duct-cannulated (BDC) rats allowed the elucidation of in vivo clearance pathways. Distribution kinetics was evaluated in the tissues and tumors of mice to support PK-PD correlation. In vitro studies enabled the evaluation of cell uptake mechanisms and cell retention properties.

**Results** Here, we show that A947 quickly distributes into rat tissues after IV dosing, where it accumulates and is retained in tissues such as the lung and liver although it undergoes fast clearance from circulation. Similar uptake/retention kinetics enable tumor growth inhibition over 2–3 weeks in a lung cancer model. A947 quickly excretes in the bile of rats. Solute carrier (SLC) transporters are involved in hepatocyte uptake of PROTACs. Sustained BRM protein degradation is seen after extensive washout that supports prolonged cell retention of A947 in NCI-H1944 cells. A947 tissue exposure and pharmacodynamics are inversely correlated in tumors.

**Conclusions** Plasma sampling for VHL-PROTAC does not represent the tissue concentrations necessary for efficacy. Understanding of tissue uptake and retention could enable less frequent IV administration to be used for therapeutic dosing.

## Plain language summary

Proteolysis-targeting chimeras (PROTACs) are a type of potential cancer medicine designed to target proteins primarily present in tumours. There is limited data on how it is absorbed, distributed, metabolised and excreted from tissues. Here, we studied the tissue distribution of synthetic PROTAC molecules labelled with radioactivity following intravenous injection in rodent models. We find that PROTAC can rapidly distribute to target tumour tissues and its prolonged retention within the tumour cells can contribute to prevention of further tumour growth, as demonstrated in the lung cancer model. These findings suggest the evaluation of PROTAC therapeutic effectiveness directly from tumour tissues provides more relevant assessment than sampling from blood circulation, which may have implications for a reduction in intravenous dosing.

Targeted protein degradation (TPD) represents a potential novel therapeutic modality with multiple degraders currently being evaluated in clinical trials[1,2]. Bi-functional protein degraders, also broadly known as proteolysis-targeting chimeras (PROTACs), consist of two small molecule ligands joined by a linker in which one ligand binds a protein of interest (POI) while the other recruits and binds an E3 ubiquitin ligase. Common E3 ligases include cereblon (CRBN), the von Hippel-Lindau tumor suppressor (VHL), and X-linked inhibitor of apoptosis protein (XIAP). The ubiquitination of the target protein is then recognized by the proteasome for

degradation[3,4]. Thus, the event-driven pharmacology of a bi-functional degrader distinguishes PROTACs from classical inhibitors that are required to maintain certain concentrations for modulating target activities.

PROTACs are large and flexible molecules, and their physicochemical properties typically include high molecular weight (MW), high lipophilicity, high polar surface area (PSA), high number of rotatable bonds, poor aqueous solubility, and limited cellular permeability. These properties place bi-functional PROTACs beyond the Lipinski's rule-of-5 (bRo5) space[5–8]. Measurement of many in vitro ADME and physicochemical properties like

[1]Genentech; 1 DNA Way, South San Francisco, CA 94080, USA. [2]Arvinas; 5 Science Park, 395 Winchester Ave, New Haven, CT 06511, USA. ✉e-mail: zhang.donglu@gene.com; hop.cornelis@gene.com

permeability, lipophilicity, pKa value(s), and protein binding is challenging for bi-functional degraders due to their non-specific binding and limited solubility[9–11]. In vitro metabolic stability for degraders often does not predict in vivo clearance[12]. Additionally, in vitro cell-based permeability measurements tend to be low although the molecule may display potent activity in the cell. At the same time, low in vitro permeability and solubility are not often predictive of in vivo absorption[13]. Due to the high MW and flexible structures of PROTACs, their permeability is difficult to improve without affecting cell potency and selectivity[11,14,15]. The limited systemic exposure following oral administration and low exposures following intravenous (IV) administration that has been observed for several VHL-based bi-functional degraders suggests that this class of degraders are characterized by limited oral absorption and high clearance[16,17]. For example, CRBN-based degraders, such as Bavdegalutamide (ARV-110) and Vepdegestrant (ARV-471), were orally dosed in the clinic with little detectable in vitro permeability[4,18]. Consequently, the development of orally bioavailable degraders has been challenging. Despite this challenge, no formal Drug Metabolism and Pharmacokinetics (DMPK) strategies are in place to optimize these molecules for drug-like PK properties due to the low predictive power of in vitro tools[19]. Therefore, poor physicochemical properties have limited effective optimization of in vitro and in vivo PK profiles of PROTACs. With the poor oral bioavailability observed for VHL-containing PROTACs, the following three questions will allow us to strategically design absorption, distribution, metabolism and elimination (ADME) studies and project support: (1) Will these molecules show selective tissue uptake despite fast plasma clearance? (2) What is the in vivo route of clearance (metabolism or elimination of unchanged drug)? (3) Based on tissue retention and not the plasma pharmacokinetic (PK) profile, can the dosing regimen be further optimized to support efficacy?

The switch/sucrose non-fermentable (SWI/SNF) protein complex facilitates the remodeling of chromatin to effect transcriptional regulation and DNA repair[20,21]. SMARCA2 (BRM) and SMARCA4 are the ATPases of the SWI/SNF chromatin remodeling complex. SMARCA2 has become an appealing synthetic-lethal therapeutic target for various cancers, including non-small cell lung cancer (NSCLC), as mutational loss of *SMARCA4* in many cancers leads to a functional dependency on residual *SMARCA2* activity [22–25]. Recently, a potent and selective SMARCA2 PROTAC molecule, A947, was reported with selective target degradation in the absence of selective PROTAC binding [26].

The physicochemical properties of A947 pose similar fundamental challenges as other VHL-based hetero bi-functional PROTACs. In this regard, A947 has a MW of 1121.4 Da and a TPSA of 224 angstrom$^2$ with multiple pKa values of 9.84, 6.57, 4.52, and 3.20. To answer basic ADME questions, we design experiments to directly investigate tissue uptake and retention and associated mechanisms. To this end, synthesis of $^{14}$C-A947 enables studying tissue distribution kinetics by QWBA in rats and tissue excision following IV administration of this tool compound. QWBA analysis at different time points allows assessment of tissue uptake and retention kinetics in all tissues. Tissue excision allows the profiling of radioactivity in selected tissues. In addition, we also determine the protein binding of A947 in plasma and tissues with close monitoring of the radioactivity mass balance at each step. Furthermore, BDC rats allow elucidation of in vivo metabolic stability and clearance pathways. The VHL-based PROTAC A947 does localize into tissues, and once the compound is in, the elimination of A947 from tissues is slow. These properties may be desirable because sustained exposure in tissues and tumours can promote the growth inhibition of tumours in the intended organs, such as the lung and liver. Furthermore, prolonged retention can enable less frequent dosing through IV administration. In vitro studies further investigate the tissue uptake mechanism and cell retention properties as well as drug-drug interaction (DDI) potential.

## Methods
### Materials
The radiolabel $^{14}$C-A947 was synthesized, and the synthesis procedure is described in the supplementary method for Synthesis of radiolabel.

The material had a radiospecific activity of 55 mCi/mmol (43.3 μCi/mg), chemical purity of 98.2%, and radiopurity of 97.2%. The material was stored at −70 °C, and the radiochemical purity was verified by HPLC analysis before use. The VHL ligand A2702 was synthesized in-house. Cell lines were obtained from the American Type Culture Collection (ATCC). Chemicals and reagents were purchased from Sigma-Aldrich unless specified.

### In vivo rat tissue distribution study design
All animal studies complied with the ethical regulations and humane endpoint criteria according to the NIH Guidelines for the Care and Use of Laboratory Animals. All procedures were approved by and conformed to the guidelines and principles set by the Institutional Animal Care and Use Committee and were carried out in an Association for the Assessment and Accreditation of Laboratory Animal Care (AAALAC)-accredited facility.

Male Sprague Dawley rats, intact or bile-cannulated, were from Envigo RMS, Inc. The animals were acclimated to study conditions for five days prior to dose administration. During acclimation, animals were individually housed in suspended, stainless steel, wire-mesh cages. During the test period, animals were placed in Nalgene cages designed for the collection of urine, feces, and bile. Certified Rodent Diet #2016C or 2016CM (Envigo RMS, Inc.) were provided ad libitum. Water was provided fresh daily. Environmental controls for the animal room were set to maintain a temperature of 20–26 °C, a relative humidity of 50 ± 20%, and a 12-hour light/12-hour dark cycle. At dosing, animals weighed 229–331 g and were 7–12 weeks of age.

Female CB17/Icr-*Prkdcscid*/IcrIcoCrl (Fox Chase CB17) mice aged 6–8 weeks were purchased from Charles River laboratories. Mice were housed in individually ventilated cages within animal rooms maintained on a 14:10 h, light:dark cycle. Animal rooms were temperature controlled between 20 and 26 °C, and humidity-controlled 30–70% with 10–15 room air exchanges per hour. Mice received food and water *ad libitum* and were allowed to acclimate for 1–2 weeks before being used for experiments. Genentech is an AAALAC-accredited facility and all animal activities in the research studies were conducted under protocols approved by the Genentech Institutional Animal Care and Use Committee (IACUC).

Three BDC rats were used for collection of bile, urine, and feces, five intact rats for collection of plasma, blood, urine, feces, and tissues, and eight intact rats for collection of blood and carcasses for QWBA. For BDC rats, a solution of taurocholic acid (2.3 mg/mL in 0.9% saline) was infused via the distal (duodenal) cannula at a rate of 0.9 mL/hour until the time of sacrifice.

The vehicle for the IV dose was 20% 2-hydroxypropyl-β-cyclodextrin (HP-β-CD) in 50 mM sodium acetate buffer, pH 4.0. On the day before dosing, 24.042 mg of $^{14}$C-A947 was weighed, and 6 mL of vehicle was added. The dose formulation was magnetically stirred overnight at ambient temperature. On the day of dosing, the dose formulation was centrifuged for 15 minutes at ~4000 × *g* at ambient temperature. The supernatant was removed, and the formulation appeared to be a very slight yellow solution. Stability was checked by HPLC analysis of pre- and post-dose aliquots. Animals were not fasted. The actual amount administered was determined by weighing the dose syringe before and after dose administration. The IV dose of 4 mg/kg was administered via a tail vein.

### Rat in vivo sample collection
From BDC rats, urine, feces, and bile samples were collected from animals predose, at 0–8, 8–24 h, and at 24-hour intervals through 168 h. Urine was collected in plastic containers surrounded by dry ice. The cages were rinsed at the end of the study with water. The carcasses were collected. Before the carcasses were frozen for radioanalysis, the esophagus, stomach, small intestine, and large intestine/cecum were excised, and rinsed with saline. The GI tract contents were analyzed separately.

For QWBA rats, one animal/time point was prepared for QWBA at approximately 0.25, 1, 4, 8, 24, 48, 168, and 336 hours. Animals were sacrificed via exsanguination (cardiac puncture) under isoflurane anesthesia, and blood (~2–10 mL) was collected into tubes containing $K_2$EDTA anticoagulant prior to collection of carcasses for QWBA. Blood was

maintained on wet ice until being aliquoted for radioanalysis and centrifuged to obtain plasma. Immediately after blood collection, the carcasses were frozen in a hexane/dry ice bath for ~8 min. Each carcass was drained, blotted dry, placed into an appropriately labeled bag, and placed on dry ice or stored at ~−70 °C for at least 2 hours. The frozen carcasses were embedded in chilled carboxymethylcellulose and frozen into blocks. Embedded carcasses were stored at ~−20 °C in preparation for autoradiographic analysis. The samples were stored at −70 °C before analysis. The remaining blood was centrifuged at $1700 \times g$ for 10 min at ~5 °C.

For tissue excision intact rats, one animal/time point was sacrificed by exsanguination (cardiac puncture) under isoflurane anesthesia at 2, 24, 48, 168, and 336 hours. Blood (~2–10 mL) was collected into tubes containing di-potassium ethylenediaminetetraacetic acid ($K_2EDTA$) anticoagulant from all animals at sacrifice. Samples were maintained on wet ice until centrifuged to obtain plasma.

The following matrices were collected from one animal per time point: Adrenal glands (left and right, analyzed separately), bone (femur), bone marrow (femur), brain, esophagus, esophageal contents, eyes (left and right, analyzed separately), fat (reproductive), heart, kidneys (left and right, analyzed separately), large intestine (including cecum), large intestine and cecum contents, liver, lungs, lymph nodes (mesenteric), muscle (thigh), pancreas, prostate gland, salivary gland, skin (dorsal, shaved), small intestine, small intestinal contents, spleen, stomach, stomach contents, testes (left and right, analyzed separately), thymus, thyroid, and urinary bladder. Tissues were excised, rinsed with saline and blotted dry, as appropriate, weighed, and placed on dry ice prior to storage at ~−70 °C. The residual carcass was saved for radioanalysis.

For intact rats, urine and feces were collected at 24-hour intervals through 168 hours and 336 hours from three animals. The carcasses were saved for radioanalysis.

## QWBA methodology
Prior to section collection, standards fortified with [14]C radioactivity were placed into the frozen block containing the carcass and were used for monitoring the uniformity of section thickness. Appropriate sections were collected on adhesive tape at 40 μm thickness, in a Leica CM 3600 cryomicrotome in accordance with Covance standard operating procedures. Sections were collected at five levels of interest in the sagittal plane. All major tissues, organs, and biological fluids were represented. Collected sections were dried at ~−20 °C. A section set from each animal was prepared by mounting a representative section from each level of interest. Mounted sections were tightly wrapped with Mylar film and exposed on phosphor-imaging screens along with fortified blood standards for subsequent calibration of the image analysis software. Screens were exposed for 4 days. Exposed screens were scanned using a Typhoon scanner. The autoradiographic standard image data were sampled using InterFocus Imaging Ltd. MCID™ Analysis software to create a calibrated standard curve. Specified tissues, organs, and fluids were analyzed. Tissue concentrations were interpolated from each standard curve as nano-curies/g (the blood calibration standards were approximately 2300, 750, 250, 75, 25, 5, and 1.5 nCi/g) and then converted to ng equivalents/g on the basis of the test article-specific activity. Tissue concentration data are presented in tabular format. Autoradiographs were annotated.

## Rat sample preparation and radioanalysis
All samples were analyzed for radioactivity in PerkinElmer Model 2910TR liquid scintillation counters for at least 5 min or 100,000 counts. Each sample was homogenized or mixed, as applicable, before radioanalysis in duplicate within 10% of the mean value. Scintillation counting data were automatically corrected for counting efficiency using the external standardization technique and an instrument-stored quench curve generated from a series of sealed quenched standards. Blood was solubilized in Soluene for at least 1 hour at 60 °C. The samples were allowed to sit at least overnight to allow any foaming to dissipate. Ultima Gold XR scintillation cocktail was added, and the samples were shaken and analyzed by liquid scintillation

counting (LSC). Plasma was also analyzed by LSC. The urine, bile, cage rinse, and cage wash samples were mixed by shaking, and duplicate weighed aliquots were analyzed directly by LSC.

Water and ethanol (1:1 v:v) was added to feces for homogenization at 1:5 (feces:water) with a probe-type homogenizer. The homogenate was digested in 1 N sodium hydroxide in an oven set to 60 °C until dissolved.

Each carcass was digested in a weighed amount of 1 N sodium hydroxide until dissolved. Ethanol was added, and the sample was homogenized by mixing.

Adrenal glands, bone marrow, eyes, thymus, thyroid, urinary bladder, brain, heart, kidneys, liver, lungs, lymph nodes, pancreas, prostate gland, salivary gland, spleen, testes, and other tissues were homogenized in water and ethanol (1:1 v:v) at approximate solvent:tissue ratio between 3:1 and 4:1.

## Preparation of rat samples for metabolite profiling
Plasma samples at 2, 24, and 48 hours were extracted twice with 3–4 volumes of acetonitrile, sonicated, vortex mixed, centrifuged, and the supernatants were removed. The extraction was repeated with 3–4 volumes of methanol. The recoveries were 58.8–72.9% from the first extraction and additional 4–10% from the 2nd extraction. Total extraction recovery was 68.2 to 77.1%. The supernatants were combined and evaporated to dryness under nitrogen at ambient temperature and reconstituted in MeOH:water (1:2, v:v, 300 μL). The reconstitution recoveries were 78.2–85.2%.

Tissue homogenate samples at 24 hours were extracted as plasma samples. The extraction recoveries ranged from 59.3 to 111%. Urine samples were pooled (5% w/v) across time intervals for each animal and further pooled across animals to generate a 0–168-hour pool. The pooled urine samples containing 10% MeOH were analyzed by liquid chromatography-high resolution mass spectrometry (LC-HRMS).

Bile samples were pooled (5% w/v) across time intervals and further pooled across animals to generate a 0–168-hour pool. The pooled bile sample containing 10% MeOH was analyzed by LC-HRMS.

Feces samples obtained from BDC and intact rats were pooled separately across time intervals for each animal (5% w/v) and further pooled across animals for each group to generate a 0–68 hour pooled sample. Approximately 1–2 g of each pooled feces sample was extracted as the same protocol used for plasma samples. Total extracted recovery was 41.9–56.6%. The reconstituted samples were analyzed by LC-HRMS.

The reconstituted samples were analyzed by LC-HRMS, with eluent fractions collected at 10-second intervals into 96-well plates containing solid scintillant. Radioactivity in each well was determined using TopCount analysis, and radiochemical profiles were generated based on radioactivity counts. Structural elucidation was performed based on mass spectral interpretation to identify metabolites.

The extraction pellets from plasma, feces and tissue homogenates were further extracted with 1 N sodium hydroxide (3 mL), vortex mixed, and placed in an oven pre-heated to 40 °C until residues had been solubilized. For plasma, solubilization recoveries were 19.5% to 33.6%. Total recovery was 96.6% to 103%. For feces, solubilization recoveries ranged from 43.4% to 55.3%. Total recovery was 97.2% to 100%. For tissues, Solubilization recoveries ranged from 2.57% to 31.6%. Total recovery was 74.3% to 122%.

For metabolite profiling and identification, LC-MS analysis used a Shimadzu/Nexera LC-30 AD liquid chromatographic system and a ThermoFisher Scientific Q Exactive with a positive/negative heated electrospray interface. The column was a Waters XSelect HSS T3 4.6 mm × 250 mm, 3.5 μm and maintained at 40 °C. The mobile phases were (A) 20 mM ammonium acetate in water and (B) acetonitrile. The gradient started with 15% B, which was maintained for 3 min, increased from 15% to 80% B from 3 to 35 min, and 80% to 95% B from 35 to 40 min, kept at 95% B for 3 min, then decreased 95% to 15% B from 43 to 43.5 min, then re-equilibrated at 15% B from 43.5–55 min. The flow was 1 mL/min and split 20% to mass analysis and 80% collected for radioactivity counting. Mass spectrometry data was acquired with full scans at a resolution of 70,000 for a m/z range of 300 to 1500, and MS/MS scans were acquired at a resolution of 17,500. The mass spectrometer was operated with a normalized collision energy of 30

(ESI+), S-Lens RF level of 60, source voltage of +3.5 kV, Capillary temperature of 320 °C, and Probe heater temperature of 300 °C.

## Protein binding measurement

Control plasma was purchased from BioIVT, Hicksville, New York. Pooled plasma was obtained from at least three rats with $K_2EDTA$ as the anticoagulant. Control plasma was used to test the suitability of the system and for positive control analysis. Dulbecco's phosphate-buffered saline (DPBS) (Millipore Sigma, St Louis, Missouri) was used to test the suitability of the system and as the receiver side of the dialysis device. Tissue homogenate samples (lungs, spleen at 24 hours) and plasma (4, 24, and 48 hours) were used for tissue protein binding. $^{14}C$-A947 or $^{14}C$-warfarin (positive control) were dissolved in acetonitrile. The concentration of acetonitrile did not exceed 1% of the final volume. The stock solution was added to the matrix (plasma or DPBS) and swirled to mix. The fortified matrix was then incubated at 37 °C for 15 min. Triplicate aliquots of the fortified matrix were analyzed by LSC.

The high throughput dialysis apparatus, model HTD96b, was used (HTDialysis LLC, Gales Ferry, Connecticut). Prior to assembly, dialysis membrane strips (molecular weight cutoff of 6–8 kDa) were hydrated, and the Teflon bars were assembled according to manufacturer's instructions, with dialysis membrane strips laid between bars creating two compartments per well. The assembled unit was locked in place in the steel base plate. A 150 μL aliquot of each fortified matrix (tissue homogenates, plasma, or DPBS) and DPBS were added to the donor and receiver side of the HTD wells in a Teflon plate, respectively, and the plate was sealed. Samples were incubated at 37 °C and rotated at 300 rpm for 7 hours. After incubation, the seal was removed, and tissue homogenates, plasma, or DPBS and dialysate from each device insert were analyzed by LSC. All protein binding determinations were performed in triplicate.

Protein binding of $^{14}C$-A947 in homogenates of rat lungs (24 hours), and other tissue (24 hours), plasma (4, 24, and 48 hours), and control plasma was conducted (25,000 dpm/mL). Dialysis was performed accordingly. The radioactivity concentration of the test article in the donor and receiver samples was determined by LSC (count time 60 min) for the calculation of protein binding. To test non-specific binding, $^{14}C$-A947 fortified DPBS and plasma (at a final radioactivity concentration of 25,000 dpm/mL) were added to the donor side of the HTD device and dialysis was performed accordingly. After the designated time, aliquots of the individual donor and receiver samples were removed and analyzed by LSC. The remaining sample volume was stored at −20 °C.

In the positive controls, protein binding of warfarin in rat control plasma was conducted. The final warfarin concentration in plasma was 3000 ng/mL. Dialysis was performed accordingly. The concentration of warfarin in control plasma donor and receiver samples was determined by liquid chromatographic-tandem mass spectrometry (LC-MS/MS) for calculation of protein binding.

## Mouse pharmacokinetics determination

For the PK study, a total of 42 male CD-1 mice weighing ~20–35 g were acquired from Lingchang/Vital River Laboratory Animal Co., Ltd. (Shanghai/Beijing, P.R. China) and divided into 2 treatment groups. A947 was dosed with one low dose (1 mg/kg) and one high dose group (20 mg/kg). For the low-dose groups, nine animals were required for organ quantification with termination at 24 hours. For the high-dose groups, 12 animals were required for organ quantification with termination at 48 hours. The animal was administered either 1 mg/kg or 20 mg/kg of formulation solution (10% HP-β-CD in acetate buffer at pH 4.0) by IV bolus through the left lateral tail vein injection. Blood samples (serial collection) were collected at the following time points: 0.083, 0.25, 0.5, and 1, 2, 4, 8, 24 and 48 hours (high dose group only). Tissues (lung, liver and kidney) were collected via sparse sampling at 4, 8, 12, 24, and 48 hours (high dose group only). All blood samples were collected from the saphenous vein (serial collection) or cardiac puncture (terminal time points) for each mouse and placed in tubes containing $K_2EDTA$. Blood samples were placed on wet ice, mixed, then

frozen and stored in a freezer set to −60 °C or lower until analysis. After the blood samples were collected at the terminal time points, the animals were sacrificed, and the liver, lung and kidney were harvested. The tissues were rinsed in cold water and then blotted on filter papers, weighed and homogenized using pre-cooled water at the ratio of 1:4. The tissue homogenate samples were kept at −70 °C until analysis.

For analysis, 25 μL of blood or tissue homogenate was aliquoted into a sample plate. Working standard solutions were spiked into 25 μL of blank matrix on the sample plate. An aliquot of 400 μL of internal standard solution (10 ng/mL loperamide) in acetonitrile was added to each well for protein precipitation, after which the plate was mixed and centrifuged. The supernatant (20 μL) was transferred into the MS injection plate, then diluted with water before injection for LC-MS/MS analysis.

The concentrations of A947 were quantified by an LC-MS/MS method. The LC-MS/MS system was a Waters Acquity UPLC System (Milford, MA) coupled to a Sciex Triple Quad 6500 Plus mass spectrometer (AB Sciex, Foster City, CA). The aqueous mobile phase for the analysis was 0.1% formic acid and 2 mM ammonium formate in water/ACN (v:v, 95:5) (A), and the organic mobile phase was 0.1% formic acid and 2 mM ammonium formate in ACN/water (v:v, 95:5) (B). Chromatographic separations were achieved with a Waters ACQUITY UPLC BEH C18 column (1.7 μm 2.1 × 50 mm). Column temperature was set at 50 °C. The gradient was maintained at 10% B for 0.2 min, then ramped up from 10% B to 95% B from 0.2–0.9 min and maintained at 95% B from 0.9 to 1.4 min and then re-equilibrated at 10% B from 1.41-1.6 min. The flow rate was set at 0.7 mL/min. The multiple reaction monitoring (MRM) transitions were m/z 1121.8 to 391.2 for A947 (DP 50 V, CE 75 V, CXP 11 V), and m/z 477.1 to 266.2 for loperamide as the internal standard (DP 80 V, CE 18 V, CXP 11 V). The lower limit of quantitation in all matrices was 5.1 ng/mL.

## Tumor pharmacokinetics determination

Tumor mouse PK was determined in the WT NSCLC xenograft model Calu-6. Female C.B-17 SCID (Inbred) mice that weighed 16–20 g were inoculated with 50 million Calu-6 cells in 200 μL. The tumor growth was monitored daily, and tumors were measured twice a week using digital calipers. Tumor volume was determined using the following formula (width × width × length/2), where all measurements are in mm, and the tumor volume is in $mm^3$. The treatment started once the average tumor volume reached 150–200 $mm^3$ ~3 weeks after cell implantation. The animals were treated with vehicle or A947. A947 was dosed at 10 mg/kg into the lateral tail vein intravenously. A947 was formulated for intravenous dosing in 10% 2-hydroxypropyl-β-cyclodextrin (HP-β-CD) and 50 mM sodium acetate in water (pH 4.0). All dosing solutions were filtered prior to injection using a 0.2-micron filter to ensure sterility. Mice were euthanized using an IACUC-approved method of euthanasia at 48, 96, and 168 hours. Blood and tumors were collected for storage at −80 °C after being flash-frozen in liquid nitrogen prior to analysis.

Omni International 19-645 Replacement 1.4 mm Ceramic beads (Catalog #19-645-3) were added to tissue tubes with ~0.5–1 tissue volume, and water was added to tumors at ~4 times the sample volume. The samples were homogenized with Omni Bead Ruptor homogenizer at strength 3.6 for three 45 second intervals. The standard stock solution was made in DMSO at 1 and 0.1 mg/mL. An aliquot of 100 μL of 1 mg/mL DMSO into STD 9 to STD 1 and 3 QC tubes, and 150 μL of 0.1 mg/mL substock was aliquoted into the STD 10 tube. The 3× serial dilutions were made from STD 10 to STD 1 by transferring 50 μL. The final STD concentrations ranged from 1 to 10,000 ng/mL. QC concentrations were 1800, 160, and 15 ng/mL. The homogenates (25 μL) were protein precipitated with 200 μL of acetonitrile containing internal standard loperamide (10 ng/mL) in a 96-well plate. Samples were vortexed and centrifuged at 2865 × g for 10 min. The 100 μL supernatant was transferred to a different plate and diluted with water before the samples were analyzed by LC-MS/MS with a 5 μL injection. The samples were analyzed on a Kinetex XB-C18 or phenyl-hexyl column (30 × 2.1 mm, 2.6 μm) that was eluted with a 5-95% B gradient of (A) 0.1% formic acid in $H_2O$ and (B) 0.1% formic acid in acetonitrile at 1.2 mL/min.

Quantitative analysis was performed on an AB Sciex API 4000 LC-MS/MS system coupled with a Shimadzu Prominence HPLC for analysis. The flow rate, source temperature, and curtain gas were adjusted and optimized for the system. MRM of loperamide was m/z 477.1→266.2.

## Mouse efficacy study

Female C.B-17 SCID (Inbred) mice that weighed 16–20 g were inoculated with 10 million HCC2302 lung adenocarcinoma cells (suspended in a mixture of Hank's Balanced Salt Solution containing Matrigel at a 1:1 ratio) in the right flank subcutaneously. Tumors were monitored until they reached a mean tumor volume of 110–167 $mm^3$. The mean tumor volume across all three groups was $141 \pm 16.64$ $mm^3$ (mean ± SD) at the initiation of dosing (Study WF-31, $n = 10$). Mice were given 0 (Vehicle−10% HP-β-CD and 40 mM sodium acetate and 40 mM NaCl in water pH 4.0), 20, or 40 mg/kg A947 IV injection once a week (QW) for 15 days in a volume of 5 mL/kg. Tumor volumes were measured in two dimensions (length and width) using Ultra Cal-IV calipers. The tumor volume was calculated with the following formula: Tumor size $(mm^3)$ = (longer measurement × shorter measurement$^2$)/2. Animal body weights were measured using an Adventura scale. Percent weight change was calculated using the following formula: Group percent weight change = (new weight − initial weight)/ initial weight) × 100%.

In a separate study, tumors were monitored until they reached a mean tumor volume of 121–159 $mm^3$. The mean tumor volume across all three groups was $138 \pm 10.26$ $mm^3$ (mean ± SD) at the initiation of dosing (Study WF-34, $n = 10$). Mice were given a vehicle, or 40 mg/kg A947 intravenous (IV) injection once a week (QW) or every other week (Q2W) for 29 days in a volume of 5 mL/kg.

## Oil-spin hepatocyte uptake assay

Uptake studies were conducted using the oil-spin method described previously [27]. Cryopreserved primary human hepatocytes (Lot KCN, BioIVT, Hicksville, NY) were thawed at 37 °C and re-suspended in InVitroGro-HT medium (BioIVT, Hicksville, NY). The suspension was centrifuged at $50 \times g$ for 5 minutes at 4 °C, and the resulting pellet was re-suspended to 2 million cells/mL in ice-cold KHB buffer with 10% human serum (BioIVT, Hicksville, NY). The cell suspension was equilibrated to 37 °C in a water bath for 10 minutes with or without inhibitors. At the start of the uptake experiment, 37 °C KHB buffer containing test compounds with or without inhibitors was added to the hepatocyte suspension. At various time points, the uptake was terminated by removing portions of the suspension to microcentrifuge tubes containing a lower 3 mM ammonium acetate layer and an upper mineral/silicone oil layer. For the 4 °C experiment, ice-cold buffer containing test compounds with or without inhibitors was added to ice-cold hepatocyte suspension, and the mixture was incubated on ice for the duration of the experiment. The microcentrifuge tubes were immediately centrifuged at $16,000 \times g$ for 12 seconds. The tubes were then frozen on dry ice. The frozen hepatocyte pellets were cut from the tubes and transferred to −80 °C freezer until LC-MS/MS analysis. For LC-MS/MS analysis, cell pellets were disrupted by a 70/30 acetonitrile and water mixture with 50 nM propranolol as an internal standard. The mixture was shaken for 15 minutes at room temperature before centrifugation at $4450 \times g$ at 4 °C for 15 minutes. The supernatant was transferred for LC-MS/MS injections.

## SMARCA2 re-synthesis after drug washout

NCI-H1944 cells were plated in 96-well plates (PerkinElmer) overnight prior to treatment with A947 (10 nM) or controls for 8 h. Following four washes in complete media, cell were left untreated or treated with A2702 (10 mM) or A947 (10 nM) for 48 or 168 hours prior to fixation with 4% formaldehyde for 15 min. Plates were washed three times with PBS buffer, incubated with a blocking solution (10% FCS, 1% BSA, 0.1% Triton, 0.01% Azide, X-100 in PBS) for 1.5 h, and subsequently treated with primary antibody (Cell Signaling, 11966) diluted 1:1200 in blocking buffer overnight at 4 °C. Following washing (3×) in PBS, cells were incubated with secondary antibodies (rabbit-Alexa 488, ThermoFisher A21206, 1:1000) for 1 h at

room temperature in the dark. Hoechst H3570 (ThermoFisher H3570) at 1:5000 was added to the wells, and the plates were incubated for an additional 30 min. Plates were washed 3× in PBS and imaged on an Opera Phenix™ High Content Screening System (PerkinElmer). Using Hoechst H3570 nuclear staining as a mask, nuclear SMARCA2 mean signal intensity was quantified.

## In vitro metabolism of A947

Hepatocytes Incubation: A947 was incubated at 5 µM in cryopreserved human and male Wistar Han rat hepatocytes (BioIVT) for 3 hr at 37 °C. Hepatocyte cell count was $\sim1.1$–$1.8 \times 10^6$ cells/mL. After 3 h, samples were quenched with 3 volumes of acetonitrile, vortex mixed, and centrifuged. The supernatant (5 µL) was analyzed by LC-MS using a Kinetex EVO C18 column (1.7 µm, 2.1 × 150 mm) at 40 °C with a flow rate of 0.4 mL/min. Mobile phase A was 20 mM ammonium acetate in water, and mobile phase B was acetonitrile. The column was initially held at 5% B for 2 min, increased to 75% B over 33 minutes, and then increased to 95% B over 3 min, where it was maintained for 3 minutes before returning to 5% B over 0.5 min for re-equilibration. Mass spectrometry data was acquired using positive ion mode and data-dependent $MS^2$ acquisition.

Liver Microsome Incubation: A947 was incubated at 5 µM in 0.5 mg/mL human and rat liver microsomes (Corning/BD Gentest) for 1 hr at 37 °C in 100 mM potassium phosphate buffer (pH 7.4) with 3 mM $MgCl_2$, 25 µg/mL alamethicin, 1 mM NADPH, and 5 mM UDPGA. After 1 hr, samples were quenched with 3 volumes of acetonitrile, vortex mixed, and centrifuged. The supernatant (5 µL) was analyzed by LC-MS as described above. Buffer control incubations were also included.

## Reversible P450 inhibition and time-dependent P450 inhibition (TDI) by A947

The reversible P450 inhibition assay was conducted in triplicate on a BioCel 1200 System, V11 (Agilent, CA), as previously described by Kosaka, et al.[28] A947 or positive controls in dimethyl sulfoxide (DMSO) stock was serially diluted in phosphate buffer (100 mM, pH 7.4) containing human liver microsomes (HLM) and individual P450 isoform-specific substrates. Individual P450 specific substrates included midazolam for CYP3A4/5, dextromethorphan for CYP2D6, S-(+)-mephenytoin for CYP2C19, S-(-)-warfarin for CYP2C9, paclitaxel for CYP2C8, and tacrine for CYP1A2. P450 activity was expressed as the peak area ratio of a P450-specific metabolite formed relative to its internal standard, and inhibition percentage was calculated from P450 activity of samples relative to the solvent control. Inhibition data were processed using Prism (GraphPad Software, San Diego, CA) to calculate $IC_{50}$ values for each P450 enzyme.

The time-dependent inhibition (TDI) experiment was performed in triplicate as described in literature[29]. First, A947 was serially diluted to 0, 1, 5, 10, and 20 µM in 0.3 mg/mL HLM in phosphate buffer (pH 7.4), and two separate sets of pre-incubation were performed at 37 °C for 30 min in the presence and absence of 1 mM NADPH. Incubation samples were processed and subsequently analyzed by LC-MS/MS as described in the reversible P450 inhibition assay. For each P450 enzyme, two sigmoidal dose–response curves were obtained to calculate %AUC shifts.

## Data analyses

Radioanalysis data tables were generated by Debra, Version 6.3.5.143 (LabLogic Systems Ltd., Sheffield, United Kingdom). Debra captures data from balances and scintillation counters. Statistical analyses may include such parameters as mean and standard deviation.

The mean time concentration profile was generated for each group, and an overall AUC was then calculated with these sparse samples of rats and mice. The composite PK parameters for plasma, blood, and organ tissues were determined by non-compartmental methods using the IV-bolus input model, Phoenix™ WinNonlin, version 8.3 (Certara USA, Inc., Princeton, NJ)[30]. The following parameters were estimated whenever possible:

**Fig. 1 | Concentration-time profiles and metabolite profiles of $^{14}$C-A947 in plasma and blood in rats following IV administration (4 mg/kg and 200 μCi/kg). a** Concentration-time profile of $^{14}$C-A947 radioactivity in plasma and blood in rats (solid circle, blood; hollow reverse triangle, plasma). The mean and standard deviation (SD) are plotted with individual numerical data listed in Supplementary Table 1. **b** Metabolite radioactivity profiles of plasma at 2, 24, and 48 hours from rats following IV administration, radioactivity in counts per minute (CPM) versus HPLC retention time in minute (min).

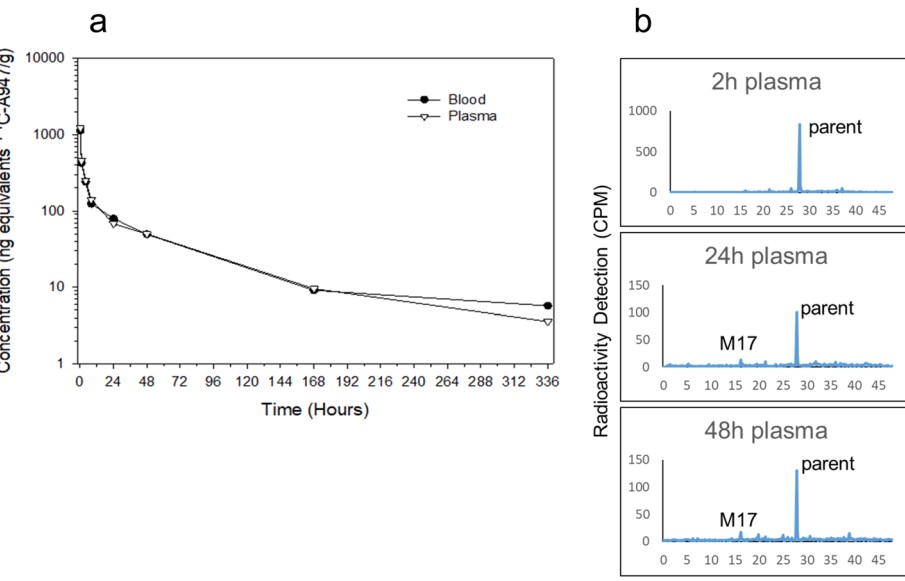

$C_0$, Back-extrapolated concentration at time 0; $C_{max}$, Maximum observed concentration; $T_{max}$, Time of maximum observed concentration; AUC$_{0-t}$, Area under the concentration-time curve from hour 0 to the last measurable concentration, estimated by the linear trapezoidal rule; AUC$_{0-336}$, Area under the concentration-time curve from hour 0 to hour 336, estimated by the linear trapezoidal rule; DN AUC$_{0-336}$, Dose normalized AUC$_{0-336}$, calculated as AUC$_{0-336}$/dose level; AUC$_{0-\infty}$, Area under the concentration-time curve from hour 0 to infinity (Day 1 only), calculated as follows: AUC$_{0-\infty}$ = AUC$_{0-t}$ + $C_t/\lambda_z$, where $C_t$ is the last measurable concentration and $\lambda_z$ is the elimination rate constant; $t_{1/2}$, Elimination half-life, determined by $\ln(2)/\lambda_z$; CL, Clearance, calculated as dose/ AUC$_{0-inf}$; $V_d$, Volume of distribution based on the terminal phase, calculated as: dose/($\lambda z \times$ AUC$_{0-\infty}$) Nominal doses and sampling times were used. Concentration values below LOQ (1.87 or 1.86 ng equivalents/g for blood or plasma, respectively) were treated as zero for descriptive statistics and kinetic analysis. Embedded zeros were excluded from kinetic analysis.

Protein binding by equilibrium dialysis:

$$\text{Percent Drug Bound} = [(Cm - Cd)/Cm] \times 100 \quad (1)$$

$$\text{Percent Drug Unbound} = 100 - \text{Percent Bound} \quad (2)$$

where: $C_d$, concentration of test article in dialysate at equilibrium; $C_m$, Concentration of test article in matrix at equilibrium.

Recovery from equilibrium dialysis:

$$\text{Recovery}\,(\%) = [(C_m \times V_m) + (C_d \times V_d)] / (C_o \times V_o) \times 100 \quad (3)$$

where: $C_m$, Concentration of test article in matrix at equilibrium; $C_d$, Concentration of test article in dialysate at equilibrium; $C_o$, Original concentration of test article in matrix prior to loading the dialysis device; $V_m$, Nominal volume of the matrix at equilibrium; $V_d$, Nominal volume of the dialysate at equilibrium; $V_o$, Nominal volume of the original matrix added to the dialysis device.

## Statistical analysis and reproducibility

Data is expressed as mean ± standard deviation (SD). Error bars in plots or graphs are standard deviations. Replicates ($n$) are independent measurements of the number of animals specified in individual experiments. Statistical analysis was performed using GraphPad Prism 8.0 Software (San Diego, CA, USA). Unpaired $T$ test was analyzed to determine the significance between each group.

## Reporting summary

Further information on research design is available in the Nature Portfolio Reporting Summary linked to this article.

## Results

### Tissue distribution in rats

Following intravenous administration of $^{14}$C-A947 at a dose of 4 mg/kg and 200 μCi/kg, the concentration-time profiles of radioactivity in blood and plasma indicated a biphasic elimination pattern (Fig. 1a, Supplementary Table 1). The initial phase was a steep decrease by greater than 90% within the first 12 hours post-administration, indicating a rapid clearance from circulation. This initial decline was followed by a slow elimination phase through 336 hours. The maximum radioactivity concentration ($C_{max}$) was identified at the first sampling time of 0.25 hours ($T_{max}$) in blood and plasma at 1.13 μg/g and 1.22 μg/g, respectively. Radioactivity was still detectable in the range of 3–6 ng/g in plasma and in blood at 336 hours post-administration. The terminal elimination half-life ($t_{1/2}$) was 72–75 hours in plasma and blood. Consistent with its half-life, the terminal clearance (CL) was low (~5.8 mL/min/kg), and the volume of distribution ($V_d$) was high (23.9 L/kg in plasma), suggesting marked bio-distribution to tissues and slow clearance from tissues (Table 1). Similar AUC$_{0-\infty}$ exposures were observed in plasma and blood (~18 μg h/g). The mean blood-to-plasma concentration ratios ranged from 0.9–1.6 through 336 hours, indicating equal blood-to-plasma partitioning. Parent was the major radioactive component with little metabolites including diglucuronide M17, which was present in plasma at 2, 24, and 48 hours (Fig. 1b).

$^{14}$C-A947 was widely distributed in tissues following IV administration with $C_{max}$ in most tissues (excluding gastrointestinal contents, bile, and urine) observed at the first sampling time of 0.25 hours, with gradually decreasing concentrations through 336 hours (Fig. 2a, b, Supplementary Fig. 1, Table 1, Supplementary Data 1–3). Similar tissue distributions were observed in tissue excision rats (Fig. 2b, Supplementary Tables 4 and 5). The tissues that showed the highest radioactivity in the QWBA group were the adrenal glands, kidneys, liver, lungs, thyroid gland, pancreas, salivary glands, and spleen with a $C_{max}$ of 17, 22, 32, 12, 18, 9, 12, and 21 μg/g, respectively (Fig. 2a, Supplementary Fig. 1a, b, Supplementary Data 1 and Supplementary Table 4). Half-life ($t_{1/2}$) ranged from 87–100 hours in the tissues, including the kidney cortex, kidney medulla, kidneys, liver, lungs, blood, muscle, and spinal cord, indicating that the rate of elimination of A947 is similar between these tissues. The radioactivity $t_{1/2}$ ranged from 11 hours in the brain cerebellum to 167 hours in the small intestine (Supplementary Data 3). Parent was the major radioactive component in multiple tissues

**Table 1 | Pharmacokinetic parameters for radioactivity in blood and tissues determined by quantitative whole-body auto-radiography from rats after intravenous administration of $^{14}$C-A947 (4 mg/kg, 200 μCi/kg)**

| Matrix | $T_{max}$ (h) | $t_{1/2}$ (h) | $C_{max}$ (µg eq/g) | AUC$_{0-t}$ (µg eq h/g) | AUC$_{0-336}$ (µg eq h/g) | AUC$_{0-\infty}$ (µg eq h/g) | DN AUC$_{0-336}$ (h*µg eq/g) |
|---|---|---|---|---|---|---|---|
| Adrenal gland(s) | 1.0 | 130 | 17.1 | 2120 | 2120 | 2620 | 530 |
| Blood | 0.25 | 97.1 | 1.23 | 18.8 | 22.1 | NR | 5.5 |
| Bone marrow | 0.25 | NR² | 4.82 | 753 | 753 | NR | 188 |
| Brain | 0.25 | 41.8 | 0.04 | 0.71 | 1.02 | NR | 0.26 |
| Brain cerebellum | 0.25 | 11.0 | 0.05 | 0.26 | 0.46 | NR | 0.12 |
| Brain cerebrum | 1.0 | NR¹ | 0.05 | 0.26 | NC | NR¹ | NC |
| Brain choroid plexus | 24.0 | NR² | 5.27 | 844 | 844 | NR | 211 |
| Brain olfactory lobe | 0.25 | NR¹ | 0.14 | 0.16 | NC | NR¹ | NC |
| Brain thalamus | 0.25 | NR¹ | 0.03 | 0.004 | NC | NR¹ | NC |
| Eye(s) | 1.0 | 117 | 0.61 | 44.2 | 44.2 | 50.1 | 11.1 |
| Heart | 0.25 | 111 | 8.8 | 273 | 273 | 307 | 68.3 |
| Kidney(s) | 0.25 | 84.6 | 22.3 | 1640 | 1640 | 1760 | 409 |
| Liver | 0.25 | 96.4 | 32 | 1270 | 1270 | 1370 | 318 |
| Lung(s) | 0.25 | 99.7 | 11.6 | 313 | 313 | 339 | 78.3 |
| Lymph node(s) | 8.0 | NR² | 2.5 | 371 | 371 | NR | 92.8 |
| Pancreas | 0.25 | 110 | 8.87 | 963 | 963 | 1080 | 241 |
| Prostate gland | 0.25 | NR² | 3.3 | 234 | 234 | NR | 58.6 |
| Salivary gland, parotid | 1.0 | NR² | 4.7 | 559 | 559 | NR | 140 |
| Salivary gland, submandibular | 8.0 | NR² | 7.6 | 1050 | 1050 | NR | 263 |
| Spleen | 0.25 | NR² | 20.6 | 2260 | 2260 | NR | 566 |
| Testis(es) | 0.25 | NR² | 0.09 | 15.5 | 15.5 | NR | 3.9 |
| Thymus | 0.25 | NR² | 0.10 | 268 | 268 | NR | 67.1 |
| Thyroid | 8.0 | NR² | 18 | 3130 | 3130 | NR | 782 |

eqEquivalents $^{14}$C-A947.

*NC* Not calculated because <3 measurable concentrations after $C_{max}$.

*NR* Not reportable as %AUC_extrap >20%.

*NR¹* Not reportable due to an inability to characterize the elimination phase.

*NR²* Not reportable as $t_{1/2}$ measured over <2× collection interval.

(adrenal, brain, bone marrow, eye, heart, kidney, liver, lung, lymph, pancreas, prostate, salivary gland, testis, and thymus) after 24 hours (Fig. 2c).

The tissue-to-plasma ratios were greater than 1 for most of the tissues at 0.25 hours and gradually increased through 336 hours, indicating a slow rate of elimination from tissues compared with plasma (Supplementary Data 2 and Supplementary Table 5). The tissues with the highest tissue-to-plasma ratios at 0.25 hours in the QWBA analysis were the adrenal glands (14), kidneys (18), liver (26), and lung (10). In contrast, the tissues that showed the highest tissue-to-plasma ratios at 336 hours were adrenal glands (758), bone marrow (374), brain choroid plexus (461), brown fat (444), kidneys (284), liver (204), spleen (949), and thyroid gland (1440, with 5.13 µg/g). A significant increase in tissue-to-plasma ratios for tissues that did not show significant partitioning at the first sampling time (bone marrow, brain choroid plexus, brown fat, lymph nodes, pituitary gland, pancreas, and salivary glands) suggest that the rates of elimination were different between these tissues.

Low concentrations in the brain (below the limit of quantitation, BLQ) by 168 hours (Supplementary Data 2) indicate poor penetration of $^{14}$C-A947 across the blood-brain barrier. In this regard, low concentrations were observed in different brain sections (with brain-section-to-plasma ratios of 0.07 at 1 hour to 0.35 at 24 hour) including the cerebellum, cerebrum, medulla, olfactory lobe, thalamus, and the concentrations reached BLQ by 48 hour by QWBA; however, sustained concentrations of ~0.4 µg/g were observed through 336 hours for the highly vascularized brain interface tissue, choroid plexus.

**Excretion, mass balance, and metabolism in rats**

Following IV administration of $^{14}$C-A947, the total radioactive dose recovered in BDC rats (0-168 hours) and intact rats (0-336 hours) was 95%

and 86%, respectively (Fig. 3a, Supplementary Table 2). Urine, feces, bile, and residual carcass accounted for 3.1%, 5.3%, 54.2%, and 32.0% of the administered dose from BDC rats, respectively, indicating that biliary excretion was the major route of elimination. In intact rats, urine, and feces accounted for respective 2.4% and 77.3% of the administered dose (Supplementary Fig. 2). Metabolite profiling indicated that the major component in bile was the parent (Fig. 3b, Supplementary Data 1). The metabolic pathways (Fig. 3c) identified include glucuronidation (M7), diglucuronidation (M17), sulfation (M1, M4), oxidation (M12, M14, M15, M16), reduction (M13), and hydrolysis (M8, M9). Linker cleavage was prominent in rats as well (M2, M3, M5, M10, M11, M18). Metabolite identification data are listed in Supplementary Table 6. More than 30% of the administered dose was found in bile during the first 24 hours in BDC rats and >50% was recovered during the first 5 days of bile collection. These results suggest that the majority of the dose was quickly eliminated as A947 during the first 24 hours following IV dosing. However, the presence of quantifiable radioactivity at the last time point in the excreta and in the residual carcass indicate a remarkable bio-distribution and prolonged duration of excretion. Taken together, these results indicate that the primary route of elimination for $^{14}$C-A947 is fecal via biliary excretion while urinary excretion represents a minor pathway (with <3% of the dose).

**Protein binding**

Protein binding of $^{14}$C-A947 in control plasma, control tissue homogenates, ex vivo plasma, and ex vivo tissue homogenates were determined using equilibrium dialysis. A positive control of $^{14}$C-warfarin was used to test non-specific binding and system suitability in these matrices. Low non-specific binding was observed with $^{14}$C-warfarin in spiked DPBS with >85%

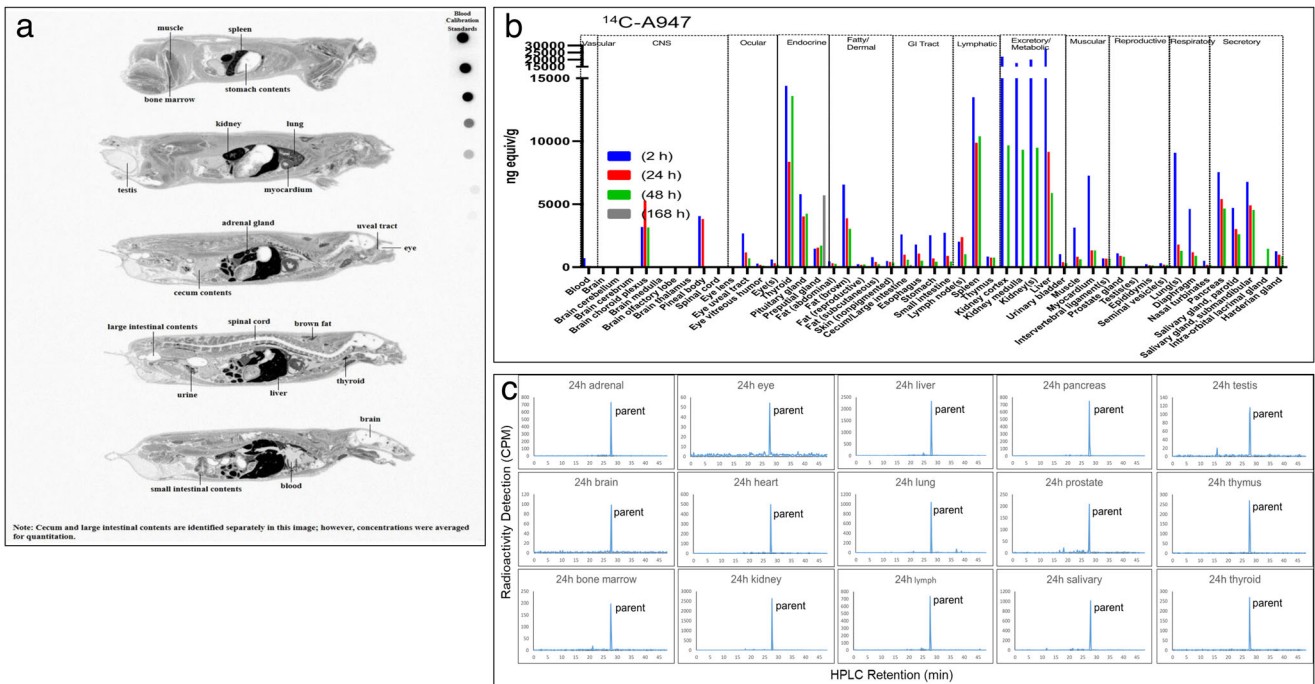

**Fig. 2 | Tissue distribution of [14]C-A947 in rats following intravenous administration (4 mg/kg and 200 µCi/kg) by quantitative whole-body autoradiograph. a** Radioactivity distribution image at 1 hour. **b** Tissue distribution of [14]C-A947 in rats by tissue excision and scintillation counting. **c** HPLC radioactivity profile of [14]C-A947 in 24-hour tissues of adrenal, brain, bone marrow, eye, heart, kidney, liver, lung, lymph, pancreas, prostate, salivary, testis, thymus, and thyroid.

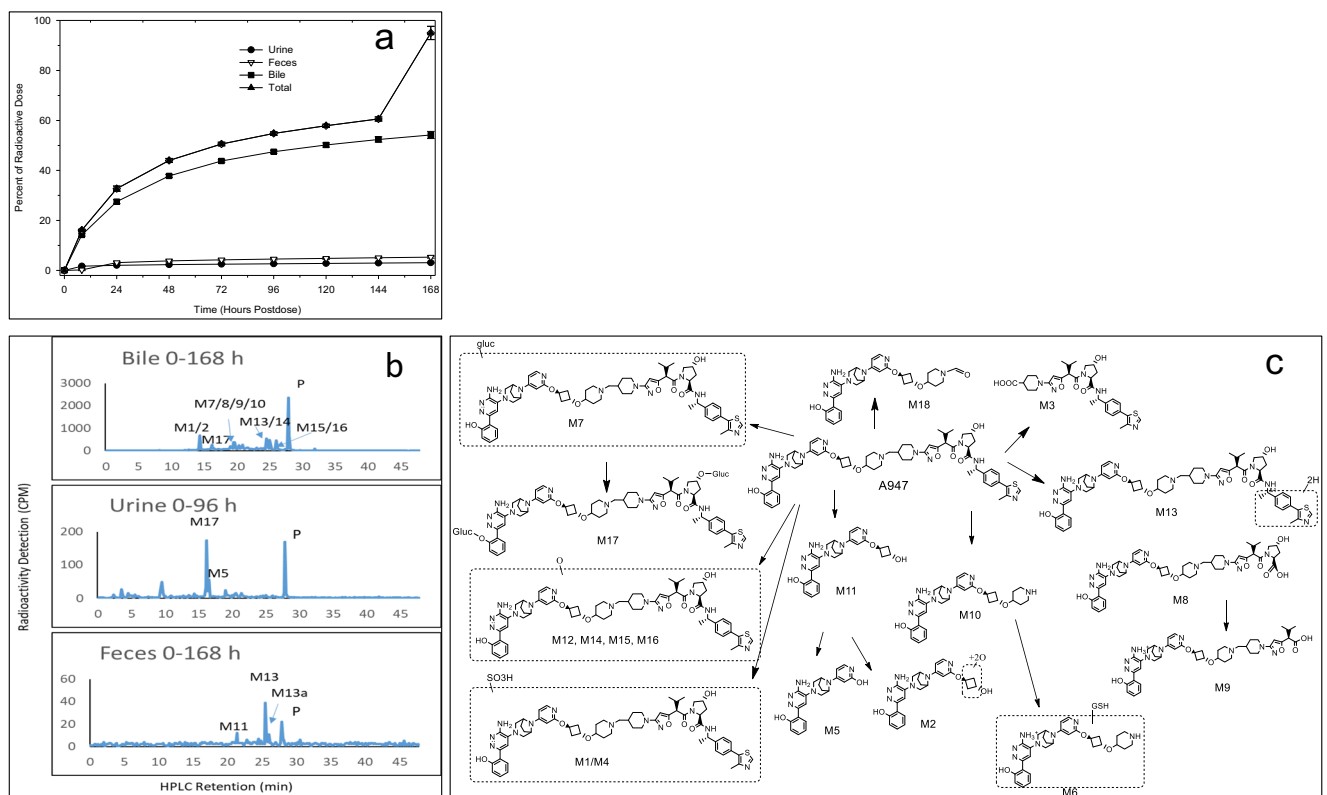

**Fig. 3 | Elimination and metabolic profiles of [14]C-A947 following intravenous administration of [14]C-A947 (4 mg/kg and 200 µCi/kg) in rats. a** Mean cumulative percent of radioactive dose in urine, bile, and feces of bile-cannulated (BDC) animals (*n* = 3) (solid circle, urine; hollow reverse triangle, feces; solid square, bile; solid triangle, total). The mean and standard deviation (SD) are plotted with individual numerical data listed in Supplementary Table 2. **b** HPLC radioactivity profiles in urine, bile and feces. **c** Structures of [14]C-A947 metabolites in urine, bile, and feces of rats.

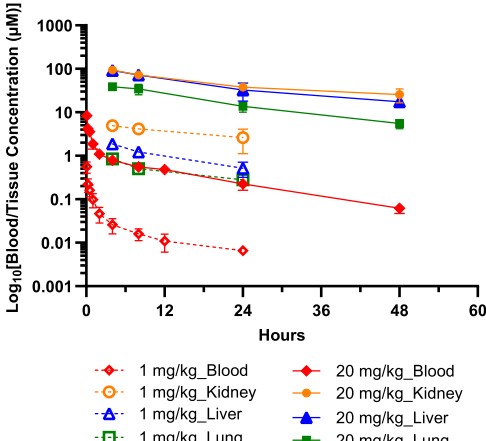

**Fig. 4 | Pharmacokinetic profiles of A947 in blood, kidney, liver, lung of CD-1 mice (n = 3) following IV administration of 1 mg/kg and 20 mg/kg.** Hollow red diamond, 1 mg/kg dose blood; solid red diamond, 20 mg/kg dose blood; hollow orange circle, 1 mg/kg dose kidney; solid orange circle, 20 mg/kg dose kidney; hollow blue triangle, 1 mg/kg dose liver; solid blue triangle, 20 mg/kg dose liver; hollow green square, 1 mg/kg dose lung; solid green square, 20 mg/kg dose lung. The mean and standard deviation (SD) are plotted with individual numerical data listed in Supplementary Data 4.

recovery. The recovery was >91% in spiked plasma, and the unbound percentage was approximately 1.5% from dialysis, as expected (Supplementary Table 7). [14]C-A947 had a <23% recovery in spiked buffer after dialysis, indicating non-specific binding. The respective bound and unbound percentage was 99.3% and 0.73% from spiked plasma after dialysis for [14]C-A947 (Supplementary Tables 8 and 9). The protein binding of [14]C-A947 from ex vivo tissues and plasma was high (Supplementary Table 9). The protein unbound percentage ranged from 0.0125% in the pancreas to 0.22% in prostate after dialysis. The unbound percentage was 0.09%, 0.0298%, and 0.04% in the lung, liver, and spleen homogenates, respectively. The varying unbound percentage after dialysis could be partially due to non-specific binding of [14]C-A947 to devices and buffer.

## Tissue distribution in mice and tumor growth inhibition in xenograft models

The A947 concentration-time profile in blood of mice after IV administration of 1 mg/kg and 20 mg/kg also indicated a biphasic elimination pattern with an initial steep decrease of >90% within initial 4 hours, indicating a rapid clearance from circulation, followed by a slow elimination phase (Fig. 4, Supplementary Data 4). Similar to rats, the compound quickly distributed to the kidney, liver, and lungs with a $T_{max}$ of 4 hours after dosing. The exposure in kidney, liver, and lung was much greater than in plasma with respective tissue-to-blood AUC ratios of 160–243, 50–127, and 26–50 for the 2 dose levels (Table 2). The terminal elimination half-life ($t_{1/2}$) was 10–15 hours in blood and tissues. Consistent with its half-life, the CL was low (~15–20 mL/min/kg) and the $V_d$ was high (20-23 L/kg in plasma) at a steady state, suggesting marked bio-distribution to tissues and slow clearance from tissues. Similar to what was observed in healthy tissues, the tumor also showed greater drug concentrations with tumor-to-blood ratios of 10, 28, and 320 at 48 hours, 96 hours, and 168 hours after IV administration of a 10 mg/kg dose in xenograft mice (Table 2). Even the estimated free drug concentrations were much higher at later time points (at 168 hours, Kp,uu could be >30). The mouse free drug concentrations in the plasma and tumor were estimated from the total A947 concentrations in mouse samples and the average free drug fractions of A947 in rat plasma and several tissues. The ratio of free drug concentrations in tissue relative to-plasma (i.e., Kp,uu) increased with time given drug clearance was slower in tissue than in plasma and tissue binding is constant. The protein binding difference in protein binding between rat and mouse tissues was not assessed. Because of tumor-

to-blood drug concentration asymmetry, A947 showed dose-dependent tumor growth inhibition in the HCC2302 xenograft model after weekly doses of 20 mg/kg or 40 mg/kg (Fig. 5, Supplementary Tables 10–13), which was accompanied by a dose-dependent pharmacodynamic effect on the *KRT80* transcript (46% vs 69% suppression, $p = 0.0036$, Table 2). Monitoring SMARCA2 protein expression by Western blotting does not provide sufficient resolution to address maximal pathway inhibition, hence we monitored expression of a transcriptional target gene (*KRT80*) that is regulated by *SMARCA2*[26]. The inverse correlation between the A947 concentrations and transcriptional target gene (*KRT80*) biomarker representing SMARCA2 levels were demonstrated in tumor tissues. In this regard, dosing every 2 weeks at 40 mg/kg still showed sustained tumor growth inhibition that was similar to 40 mg/kg dosed weekly, with comparable effects on the pharmacodynamic biomarkers SMARCA2 and *KRT80* mRNA (Table 2).

## A947 Hepatocyte uptake and cancer cell retention

The cellular uptake mechanism of A947 was investigated by measuring the cellular concentration with LC-MS/MS in hepatocytes in the presence or absence of transporter inhibitors. The compound in the cells was cleanly separated from that in the media through an oil layer by centrifugation after incubation. In the cell uptake assay, A947 showed time-dependent uptake by primary human hepatocytes for up to 3 minutes. The uptake was abolished at 4 °C, suggesting that the uptake was at least partially contributed by an active carrier-mediated process. Co-administration of 1 mM rifamycin SV largely inhibited the temperature-dependent uptake (Fig. 6a, Supplementary Table 14). Since rifamycin SV acts as a pan-solute carrier (SLC) inhibitor at 1 mM, the result suggests that SLC transporters were involved in the uptake of A947 by hepatocytes.

Cell retention of A947 was investigated through monitoring the pharmacodynamics (PD) effect of SMARCA2 degradation in cell incubations after extensive washout of the compound from the incubations in the presence or absence of competing E3 ligand. A947 degraded SMARCA2 protein in NCI-H1944 cells effectively by 8 h (by ~85%) at 10 nM (Fig. 6b). By 48 h and even 168 h following 8-hour drug treatment in NCI-H1944 cells following the drug washout, there was minimal SMARCA2 protein re-synthesized, suggesting prolonged retention of A947 in the cells. The results also suggest quick in and slow out of A947 in the cancer cells. The VHL ligand, A2702, spiked in at 1000× molar excess following washout to compete with residual A947 in cells, partially recovered the SMARCA2 levels at 48 and 168 h following washout. Additional spike-in of A947 following the washout further degraded the SMARCA2 protein.

## In vitro metabolism and DDI potential of A947

Liver metabolism of A947 was studied in incubations with hepatocytes and microsomes of rats and humans. The compound was relatively stable in these incubations, with <10% of the compound being metabolized (Supplementary Table 15, Supplementary Fig. 3). Linker cleavage was the major metabolic pathway of A947 in hepatocytes and liver microsomes, which is consistent with the major metabolic pathways of A947 in vivo in rats. In vivo, metabolism of A947 included linker cleavage and formation of glucuronides, and diglucuronide (M17) appeared to be more extensive in vivo compared to in vitro incubations, as expected. The oxidative metabolites were either not formed or were formed at a lesser extent in incubations with human liver microsomes without P450 co-factor NADPH, suggesting that the oxidative metabolism of A947 is mainly catalyzed by P450 enzymes (Supplementary Table 15). Similar metabolites were observed in rat and human hepatocytes. Chemical degradation of A947 was minimal in these incubations.

The drug interaction potential of A947 was evaluated in P450 inhibition assays[28,29]. A947 was a direct inhibitor of P450 3A with an IC50 value of ~0.54 μM with midazolam as a substrate. Interestingly, much weaker P450 3A inhibition was observed when testosterone was used as the substrate (Supplementary Table 16). A947 was also a direct inhibitor of P450 2C19 with $IC_{50}$ value of ~3 μM. A947 was also a time-dependent inhibitor of P450 3A with both midazolam and testosterone as substrates

**Table 2 | Pharmacokinetic parameters for A947 in blood, kidney, liver, and lung in CD-1 mice after intravenous administration of A947 at 1 or 20 mg/kg (a); drug concentrations in blood and tumors determined in WT NSCLC xenograft model Calu-6 following intravenous administration of A947 at 10 mg/kg (b); pharmacodynamic assessment of SMARCA2 protein levels and KRT80 mRNA levels following IV administration of A947 in xenograft models (c)**

**(a) Pharmacokinetic parameters of A947 in CD-1 mice**

| Dose (mg/kg) | Matrix | $C_{max}$ (μM) | $T_{max}$ (hour) | T1/2 (hour) | $C_{last}$ (μM) | $AUC_{last}$ (hr*μM) | $AUC_{inf}$ (hr*μM) | AUC_% extrapolated | Vd (L/kg) | CL (mL/min/kg) | Tissue:Blood AUC Ratio | AUC ratio (dose 20:1) |
|---|---|---|---|---|---|---|---|---|---|---|---|---|
| 1 | Blood | 0.56 | 0.083 | 13.1 | 0.00654 | 0.608 | 0.732 | 16.9 | 23 | 20.3 | | |
| 1 | Kidney | 4.91 | 4 | 22.6 | 2.62 | 92.8 | 178 | 47.9 | | | 243 | |
| 1 | Liver | 1.87 | 4 | 11.3 | 0.514 | 28.5 | 36.8 | 22.7 | | | 50.3 | |
| 1 | Lung | 0.843 | 4 | 13.9 | 0.28 | 13.1 | 18.8 | 30 | | | 25.6 | |
| 20 | Blood | 8.38 | 0.083 | 12.5 | 0.0622 | 19.2 | 20.3 | 5.51 | 15.8 | 14.6 | | 31.6 |
| 20 | Kidney | 94.3 | 4 | 24 | 25.5 | 2370 | 3250 | 27.1 | | | 159.6 | 25.5 |
| 20 | Liver | 91.4 | 4 | 18.5 | 17.4 | 2120 | 2580 | 18 | | | 127.2 | 74.4 |
| 20 | Lung | 38.9 | 4 | 15.2 | 5.5 | 891 | 1010 | 12 | | | 49.7 | 68 |

**(b) Drug concentration in Calu-6**

| Time points (h) | Plasma [A947] (nM) | Est. free plasma [A947] (nM) | Tumor [A947] (nM) | Est. free tumor [A947] (nM) | Est. Kp | Est. Kp,uu |
|---|---|---|---|---|---|---|
| 48 | 15.3 | 0.102 | 154 | 0.105 | 10 | 1 |
| 96 | 3.11 | 0.021 | 92 | 0.063 | 28 | 3 |
| 168 | 0.24 | 0.0016 | 77 | 0.053 | 320 | 33.1 |

**(c) SMARCA2 protein and KRT80 mRNA levels**

| Dose (mg/kg) | Route | Dosing Schedule | % SMARCA2 protein suppression | % KRT80 mRNA expression (mean) | % KRT80 mRNA expression (sd) | % KRT80 mRNA suppression | Unpaired T test |
|---|---|---|---|---|---|---|---|
| 20 | IV | QW | 97 | 54 | 17 | 46 | $p = 0.0036$ |
| 40 | IV | QW | 98 | 31 | 12 | 69 | |
| 40 | IV | Q2W | 99 | 39 | 6 | 61 | $p = 0.02$ |
| 40 | IV | QW | 99 | 30 | 8 | 70 | |

The free drug fractions in mouse plasma and tumors used the estimated average protein unbound values of rat plasma at 4, 24, and 48 hours (0.668%) and average protein unbound values of kidney, liver, lung, pancreas, prostate gland, spleen, and thymus (0.0688%) that were determined by ex vivo 24-h samples of rats following dosing radiolabeled A947 from Supplementary Table 9. Tumor xenograft protein levels of *SMARCA2* and the *SMARCA2*-dependent mRNA transcript, KRT80, were measured as previously described (Cantley et al. 2022)[26].

**Fig. 5 | Tumor growth inhibition and body weight profiles of A947 in HCC2302 xenograft models after IV injection of vehicle or A947 (*n* = 10; A947 = A10947). a, b** When tumors reached a mean tumor volume of 110–167 mm³, mice were given vehicle, 20 mg/kg or 40 mg/kg A947 IV injection once a week (QW) for 15 days in a volume of 5 mL/kg (solid circle, vehicle; solid blue square, A10947 20 mg/kg; solid red triangle, A10947 40 mg/kg). **c, d** When tumors reached a mean tumor volume of 121–159 mm³, mice were given a vehicle or 40 mg/kg A947 intravenous (IV) injection once a week (QW) or every other week (Q2W) for 29 days in a volume of 5 mL/kg (solid circle, vehicle; solid blue square, A10947 40 mg/kg Q2W; solid red triangle, A10947 40 mg/kg QW). The mean and standard deviation (SD) are plotted with individual numerical data listed in Supplementary Tables 10–13.

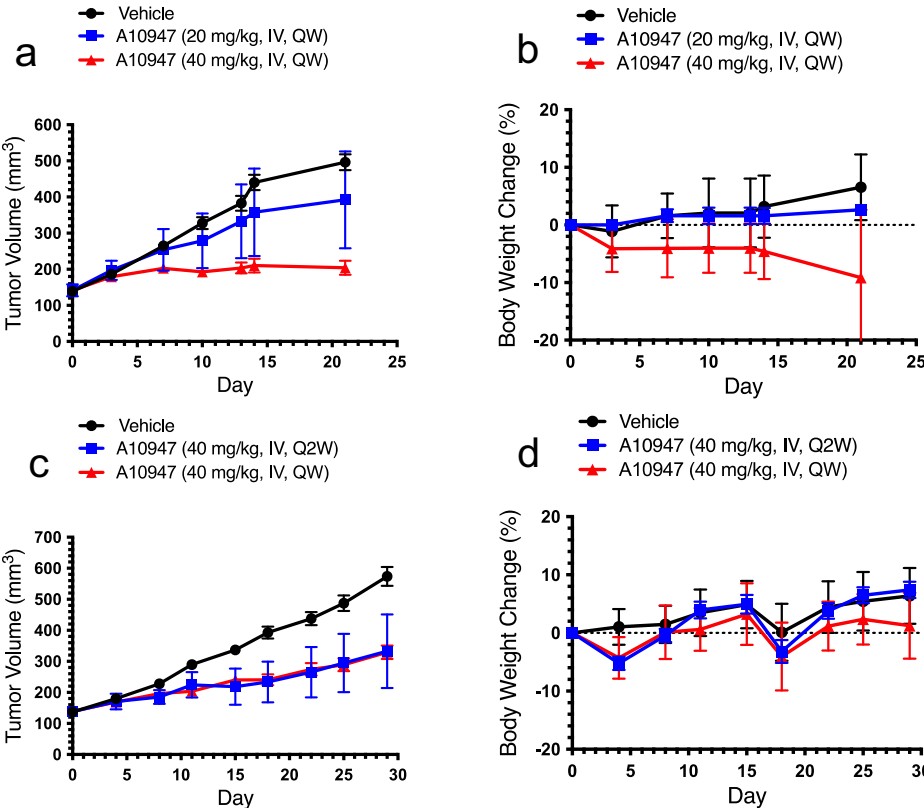

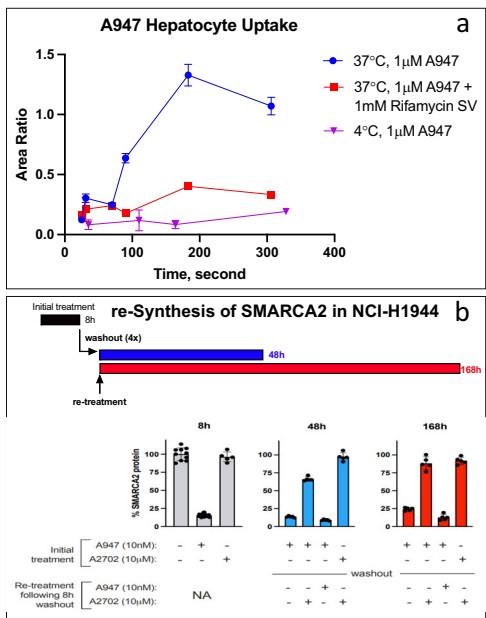

**Fig. 6 | In vitro cell uptake and retention of A947. a** Time-dependent uptake of A947 in hepatocytes in the absence and presence of pan-solute carrier transporter (SLC) inhibitor rifamycin SV ($n = 3$) (solid blue circle, 37 °C A947; solid red square, 37 °C A947+rifamycin; solid pink triangle, 4 °C A947). The cellular concentration was measured by LC/MS and represented as peak area ratios to the internal standard. Data are normalized to DMSO control treated cultures and are presented as mean ± SD from three independent replicates. Individual numerical data are listed in Supplementary Table 14). **b** Evaluation of re-synthesis of SMARCA2 protein in NCI-H1944 cells following washout of A947 ($n = 5$). Quantification of SMARCA2 protein levels in cell by immunofluorescence following treatment and washout of A947. Data are normalized to DMSO control treated cultures and are presented as mean ± SD from five independent replicates. The VHL ligand, A2702, was spiked in at 1000× molar excess following washout to compete with residual A947 in cells. SMARCA2 levels were evaluated at 48 h and 168 h following washout.

## Table 3 | Physicochemical properties of selected VHL-based PROTACs

| Compound | MW | cLogP | HBD | HBA | PSA | nRotB |
|---|---|---|---|---|---|---|
| Average ($n = 13$)[a] | 1034 (949–1159) | 6.6 (4.2–8.7) | 4.2 (3–5) | 17 (14–19) | 211 (177–233) | 26 (21–35) |
| A947 | 1121 | 8.4 | 5 | 19 | 224 | 17 |
| A005 | 1119 | 3.7 | 7 | 20 | 267 | 10 |

[a]The data are from Edmondson SD et al.[8]. The average values of each physicochemical property are listed with the range in the bracket.

(Supplementary Table 17). It is not surprising for A947 to have the P450 inhibition potential and to be a P450 substrate given the strong correlation between compound lipophilicity and interactions with P450 enzymes. A PROTAC strategy was actually used to degrade extrahepatic cytochrome P450 1B1 (CYP1B1), which is highly expressed in various tumors, for cancer prevention, therapy, and reversal of drug resistance[31].

## Discussion

This study provides the first detailed set of disposition data for a PROTAC through a comprehensive delineation of the pharmacokinetics, distribution, metabolism, and excretion of [14]C-A947 following IV administration in rats. The rat tissue distribution from QWBA analysis, along with tissue excision, indicated extensive distribution, with the highest levels measured in tissues such as the lung, liver, spleen, and kidney. Elimination of [14]C-A947 in blood and plasma indicated a biphasic pattern characterized by a rapid initial phase likely due to quick biliary excretion as well as bio-distribution followed by a prolonged terminal elimination phase. Our hepatocyte uptake experiments demonstrated that cell uptake was mediated by SLC transporters, which has not been reported as a mechanism for PROTAC cell uptake. The biliary excretion could be mediated by efflux transporter as the involvement of efflux transporter MDR1 has been implicated in the cancer cell resistance of a PROTAC[32]. The sustained exposures in selected tissues were complemented by the prolonged half-lives of ~80 hours. The drug accumulation in selective tissues was also observed in mice and in tumors from xenograft mice. The residence time of A947 in tumors, lung, liver, kidney, and other tissues appeared to be much higher than in blood or plasma, suggesting that clearance from tissues is slower compared to clearance from systemic circulation in mice. A947 is characterized by more

than a 10-fold decrease of plasma concentration during the initial 12 hours post-administration and slow elimination in peripheral organs such as lung, spleen, liver, and kidney in rodents. The sustained PD effect 1-week following washout in the cancer cells further demonstrated the cellular retention of the molecule.

Drug distribution characteristics are determined by a combination of animal physiology, such as blood flow, metabolism enzymes, transporter expression, pH, and physiochemical properties of a compound, such as permeability, solubility, pKa, lipophilicity, metabolic stability, and protein binding. Several factors can create and maintain disequilibrium between the drug concentration in plasma and tissues leading to a drug concentration asymmetry[33,34]. These factors include drug membrane permeability, uptake and efflux drug transporters, lysosomal trapping, intracellular biotransformation, pH gradients, and unique distribution properties like that of the VHL-based PROTAC, A947. A recent study suggested that the cellular uptake of some PROTACs is facilitated by interferon-induced transmembrane proteins[35]. Our hepatocyte uptake experiments support the involvement of SLC transporters in the cell uptake of a PROTAC. There are over 400 SLC transporters, and many of the SLCs with known drug substrates exhibited differential tissue expression[36]. Research efforts are underway to identify specific SLCs responsible for A947 uptake, which can help correlate the higher expression levels in tissues with greater A947 accumulation. Another VHL-containing bi-functional degrader A005 with similar physicochemical properties (Table 3) but different protein target ligand bromodomain-containing protein 4 (BRD4) also showed similar tissue distribution patterns as A947 in terms of accumulation in selected organs such as the lung, spleen, liver, and kidney and prolonged retention. These data suggest that VHL-based PROTACs have similar tissue distribution patterns due to their similar sizes and physiochemical properties. A comparison of the physicochemical properties among selected PROTACs recruiting MDM2, IAP, VHL, or CRBN E3 ligases is listed in Supplementary Table 18. Overall, these results revealed the unique tissue distribution characteristics of the VHL-based PROTACs[8]. For example, the data does not suggest that these molecules can be applied in brain disease treatment since there was low penetration to the brain and insufficient use of a drug dose[37]. Potential clinical benefits from the limited brain uptake of the Leucine-rich repeat kinase 2 (LRRK2)-targeting PROTAC, XL01126, remain to be seen[17]. Tissue distribution and clearance are two key pharmacokinetic parameters that control the in vivo pharmacological performance of a PROTAC. The sustained efficacy of A947 for 3–4 weeks after a single 40 mg/kg dose of the compound in the xenograft models suggests that the drug concentrations reached appropriate threshold levels in the tumor[38]. In addition, a tumor in the flanks of a xenograft mouse should be different from a tumor embedded in an organ such as lung or liver because a drug would need to get into the organ and tumor before inhibiting growth of the tumor embedded within the organ. The tissue uptake and prolonged retention of A947 in the lung, liver, and xenograft tumor supports the potential of A947 to inhibit growth of tumors embedded in the lung or liver despite its fast clearance from circulation.

Typical small-molecule drug discovery is often initiated with the goal of identifying active leads against a therapeutic target of interest in cell lines. Rational design or large-scale screening can quickly generate leads with high in vitro potency. While ADME optimization attempts to achieve drug

**Fig. 7 | PROTAC tissue distribution study design and analysis.** This includes quantitative whole-body autoradiograph (QWBA, the blood calibration standards were ~2300, 750, 250, 75, 25, 5, and 1.5 nCi/g), MALDI-imaging mass spectrometry (MALDI-IMS), or tissue excision for mass spectrometry, tissue and plasma drug concentration asymmetry as shown in PROTAC concentration-time profiles in tissues and plasma, and potential applications in target tissues using less frequent intravenous administration.

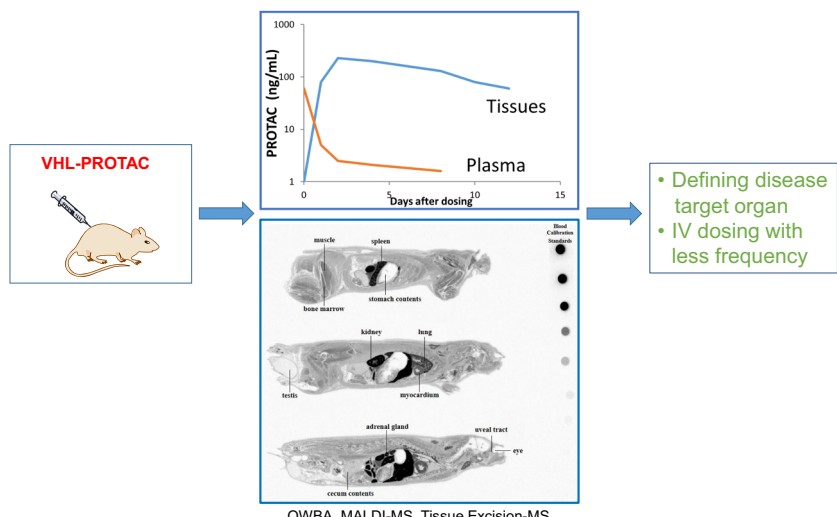

exposures in circulation following oral or IV administration in animal models, the large size and poor physiochemical properties (bRo5) of VHL-containing bi-functional degraders lead to low bioavailability or limited systemic exposure even after IV dosing. Achieving multiple DMPK parameters that are required to optimize systemic exposures of bi-functional degraders while maintaining biological potency and selectivity seems to be a difficult task. While the slow membrane permeation of PROTACs results in a drug concentration asymmetry between selective tissues and plasma, it may offer favored drug concentrations in target organs instead of the drug exposure in circulation. Our tissue accumulation data of VHL-based PROTACs provides a new drug discovery strategy to utilize the drug concentration asymmetry and supports applications of tissue-specific drug accumulation data. In addition to QWBA and tissue excision, Matrix-assisted laser desorption/ionization imaging mass spectrometry (MALDI-IMS) can also offer a method to quickly verify the selective tissue distribution of new protein degraders of therapeutic interest in a particular organ, such as the lung and liver[39]. Fig. 7 illustrates PROTAC tissue distribution study design and analysis, tissue and plasma drug concentration asymmetry, and potential applications in disease treatment through less frequent intravenous drug administration.

There are limited reports on improving permeability and oral bioavailability of VHL-based PROTACs[17,38]. There is, in general, a lack of quality data or lack of assays to collect quality data on the solubility, permeability, and physiochemical properties of PROTACS[19,40]. The liver metabolism study of A947 identified linker cleavage as a metabolic soft spot, which would point to a direction of structural modifications to improve metabolic stability and reduce a potential first-pass effect. Systemic pharmacokinetics of a VHL-based PROTAC have been much improved through a nano-particle formulation[41]. However, PD improvement was less than expected from the formulation study, which would require additional investigation to enhance tumor penetration of the PROTACs with the nano-formulations. Thus, based on the unique ADME properties of the PROTAC modality, we may be required to focus on selective tissue distribution as a way to improve PD. It is possible that the challenging ADME properties of PROTACs offer a new therapeutic opportunity for those molecules that lack the optimal ADME properties of traditional small molecules. Our results also suggest that we may need to change the criteria regarding how we utilize systemic PK data for optimization of bRo5 molecules. Novel technologies are also being developed to deliver these compounds to their targets. For example, conjugation of the BRD4- and estrogen receptor (ER)-targeting PROTACs to engineered monoclonal antibodies provides a novel means of delivering these molecules to tumors in vivo. These conjugates have enabled efficient intracellular release of the compound to support the biological activity[42–45]. Nanoparticle formulations were also developed to deliver a BRD4-targeting

PROTAC for the treatment of pancreatic cancer[46]. Now, IV dosing also offers a path forward for PROTACs since it delivers the intended dose and dose frequency that is competitive with the standard of care and is well-accepted by patients[47], which enables faster discovery of other PROTACs in general by saving time on developing unnecessary drug delivery methods.

Although oral delivery of VHL-based PROTACs is generally problematic due to their high molecular weight and poor physicochemical properties, our detailed tissue distribution study demonstrated selective tissue uptake and retention for this class of molecules following IV administration. Plasma concentrations of A947 do not represent the tissue concentrations necessary for efficacy due to fast plasma clearance, but the half-life in the selective tissues (including tumor) was sufficiently long to only require an infrequent dosing frequency (every 2 or 3 weeks) to support a sustained level of tumor growth inhibition in xenograft models. The combined properties of event-driven pharmacology with sustained tissue exposure and tissue-to-plasma concentration asymmetry of VHL-based PROTACs from less frequent IV dosing can benefit healthy organ recovery and, eventually, the overall safety profile. Overall, our data may have future clinical implications, offering a potential therapeutic option for patients harboring *SMARCA4* mutation cancers because of their efficacy and less frequent IV dosing. These findings suggest that this PROTAC and other VHL-based PROTACs, such as the previously reported VHL-based BCL-xL PROTAC by Khan et al.[48] with similar physicochemical properties can potentially sustain a long-term exposure to induce a PK-PD decoupling. These properties are highly desirable for the clinical application of these PROTACs because they allow a reduction in dosing and dosing frequency to reduce potential drug toxicity while maintaining therapeutic efficacy.

Overall, this work provides a comprehensive study of cellular uptake, distribution, metabolism and excretion of a PROTAC and a new strategy of using tissue distribution rather than the traditional plasma exposure in the discovery of a PROTAC as a potential therapeutic agent.

## List of supplementary Information
Supplementary method: synthesis of radiolabel
Supplementary Tables 1–18
Supplementary Figs. 1–4
Supplementary Data 1–4

## Data availability
The numerical data underlying Figs. 1, 3, 4, 5, and 6 are in Supplementary Tables 1, 2, Supplementary Data 4, Supplementary Tables 10–13, and 14, respectively. Mass spectral and nuclear magnetic resonance (NMR) data and spectra are in the Supplementary Methods and Supplementary Fig. 4,

respectively. Additional data is available from the corresponding author on reasonable request.

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

## Acknowledgements
We thank Philip Manteufel, Michael Fitzsimmons, Vijayabhaskar Veeravalli, Heidi Thorson from Labcorp, Michael Berlin from Arvinas, Simone Schadt from Roche, Stefan Koenig, Fabio Broccatelli, Matthew Baumgardner, Matthew O'Brien, Brian Dean, Matthew Wright, Jane Kenny, and David Stirling from Genentech for valuable discussions.

## Author contributions
Study design: D Zhang, Dragovich, Pizzano, Yauch, Khojasteh. Experimental: L Ma, Bortolon, Ye, E Chen, Y Chen, Levy. Data analysis: D Zhang, B Ma, L Ma, S Chen, E Chen, Y Chen, Liu, Pizzano, X Zhang, E Chan, Khojasteh, Yauch, Hop. Writing: D Zhang, B Ma, Dragovich, S Chen, E Chen, Khojasteh, Yauch, Hop. All authors read and approved the manuscript.

## Competing interests
The authors declare the following competing interests: D.Z., B.M., P.D., L.M., S.C., E.Chen., X.Y., J.L., E.Chan, X.Z., Y.C., E.L., R.Y., C.K., and C.H. are employees of Genentech, Roche and have stocks, and/or options of Roche. J.P. and E.B. are employees of Arvinas and have stocks, and/or options of Arvinas.
