## [Peer Review File · Communications Medicine]

Reviewers' comments:

Reviewer #1 (Remarks to the Author):

This research manuscript described the ADME of 14C-A947 and provided a new strategy that used tissue distribution rather than traditional plasma exposure to develop a PROTAC. It is interesting that though the plasma concentration of A947 is low, the tissue concentrations and long half-life in the selective tissues are sufficient for efficacy in xenograft models. Usually, drug exposure in plasma is a very important parameter to develop, but for PROTAC drugs, the tissue concentration and half-life should be considered because of their poor physical properties due to their high molecular weights. Following major comments/suggestions noted here to authors to consider answering prior to its publication in this reputed journal:

1. In Fig.3C, the structures of 14C-A947 metabolites in the urine, bile, and feces of rats are identified. The metabolites data clearly supports the readers for further optimization to facilitate better or improved PK parameters. The liver is the main metabolic organ; authors should include the liver as an organ to identify the metabolites in the liver or P450 and discuss the liver or P450 metabolites of A947.

2. In paragraph 2 of the "Result, Tissue distribution in rats" section", adrenal glands, kidneys, liver, lungs, thyroid gland, pancreas, salivary glands, and spleen with a Cmax of 17, 22, 32, 12, 18, 9, 12, and 21 µg/g, respectively (Fig. 2A, S1A, S1B, Table S1A, S1B, S1C), and the half-life ranged from 87-100 hours. Combining with Table 2 (Cmax, AUC, half-life). So, the Liver is a key target for A947, hence authors should justify the rationale for kidney cancer, but liver cancer models seem promising for these series to consider discussing the role of A947 in liver cancers.

3. A947 tissue distribution kinetics via 14C-labeled Brahma-associated protein (BRM) degrader. The data shows A947 quickly distributed to tissues after IV dosing, where it is accumulating and is retained in tissues such as the lung and spleen with tissue-to-plasma ratios of >25 at 24 hours. A947 was quickly excreted in the bile of rats. These findings provide comprehensive data for cellular uptake, distribution, metabolism, and excretion of PROTAC.

The authors used IV dosing instead of oral delivery. Authors could discuss what's the difference in metabolism and pharmacokinetics between IV and Oral delivery. Authors are also suggesting optimizing PROTACs for oral drug delivery based on metabolism and pharmacokinetics data. Hence authors should state how they proceed in oral PROTACs for the A947 series.

4. For PROTAC's half-life, except for enzymatic transformation, could authors also discuss what's another potential to undergo partial non-enzymatic degradation?

5. Given the strong association between lipophilicity and CYP inhibition, additional scrutiny of drug-drug interaction (DDI) potential is warranted for PROTACs in poorly precedented cLog P/Log D space. CYP inhibition was investigated for both 'capped' POI/E3 ligands and full PROTACs. One factor the author should be highlighted quantifying the risk of CYP inhibition.

Reviewer #2 (Remarks to the Author):

The authors present a significant body of data profiling the absorption, distribution, metabolism and excretion of an *in vivo* active SMARCA2 degrader tool compound, utilising a C14 labelled analogue of a previously published SMARCA2 degrader A947. This study is the first of its kind with respect to detailing the tissue distribution of an *i.v.* dosed VHL PROTAC and highlights very important learnings for the wider field that could prove to be of significant clinical benefit. Most crucially, the authors show that *i.v.* dosed VHL PROTACs may enable sustained efficacy in particular tissues where higher free concentration of drug is retained in comparison with plasma at long time points. There are also important learnings regarding the major routes of elimination of the molecule used here, 14C-A947. This is a very interesting and well-timed study that I would fully support for publication. It will provide an important benchmark for the field in considering when and how VHL PROTACs could be employed, particularly in oncology. A limitation of the study as presented is that it does not go on to suggest how *in vitro* adme discovery cascades could be reshaped to help with identifying and designing molecules that could capitalise on the features shown here, a challenge highlighted by the authors for current PROTAC optimisation approaches. Though this does not detract from the value of the data and analysis that is included.

I would in principle support publication of this manuscript as presented, though there are a few minor points that could be clarified to help the reader:

- On page 12 the following statement is made: 'Even the estimated free drug concentrations were much higher at later time points (168 hours, K_p , could be >30). Firstly, I think the author mean $K_{p,u}$ not K_p in this context, please correct/clarify. In addition, as the authors have used free fractions of drug in various tissues in rat *ex vivo* studies as a basis for calculating free drug in mouse tissue, it would be helpful to include a sentence or two identifying the potential limitations of that and to what extent this limits how conclusive one can be regarding the precise free concentrations in mouse tissue. It does not detract from the overarching point that if tissue binding is constant but total drug levels are changing, then the ratio of free drug concentrations in tissue relative to plasma (i.e. $K_{p,u}$) over time must be increasing.

- At the beginning of the section 'Hepatocyte uptake and cancer cell retention', the authors simply say 'in the cell uptake assay...'. Whilst I appreciate some detail is given in the methods section for this assay, a brief (one or two sentence) introduction to what this assay is and how uptake of compound was detected would significantly help reader interpretation of the following section.

Reviewer #3 (Remarks to the Author):

The authors were the first to use 14C to label PROTAC to study its tissue distribution, metabolism and clearance. Using 14C labeled A947, a von Hippel-Lindau (VHL)-based Brahma associated protein (BRM) PROTAC, they found that this PROTAC rapidly distributed into tissues but accumulated and retained in high concentrations in the thyroid, spleen, kidney and liver in rats after *iv* administration. A947 was metabolic stable and was primarily eliminated via excretion in bile. However, elimination of tissue A947 was very slow after it accumulated in tissues. Similar findings were also observed in mice. These findings suggest that this PROTAC and other VHL-based PROTACs with similar chemical and physical properties can potentially sustain a long-term exposure to induce a PK-PD decoupling as reported previously for a VHL-based BCL-xL PROTAC (Khan S et al. Nat Med, 25: 1938-1947, 2019). These properties are highly desirable for the clinical application of these PROTACs because they allow a reduction in dosing and dosing frequency to reduce potential drug toxicity while maintaining their therapeutic efficacy. In addition, using cell culture the authors also demonstrated that SLCs may be responsible for the hepatocyte uptake of A947. Identification of a specific SLC responsible for the

uptake will further advance our understanding on how PROTACs get into cells. Overall, the authors reported the first detailed study to evaluate PROTAC ADME and DMPK properties, which will produce a major impact on the future development of PROTACs and design of preclinical and clinical testing of PROTACs.

Major concerns:

1. The authors reported differential tissue distribution, accumulation, and retention of A947 after iv injection in rats using QWBA and by quantification of ¹⁴C labeled A947 in excised tissues. Even though the PROTAC appears intact chemically and should be active, its activity was not tested in tissues. The authors should measure the levels of SMARCA2 in selected tissues that had high and low A947 accumulation to see if the tissue concentrations of A947 are inversely correlated with the levels of SMARCA2.
2. In Table 4, the authors showed that 20 mg/kg A947 was equally effective as 40 mg/kg A947 in reducing the levels of SMARCA2 in tumor xenografts, but the former had no significant effect on tumor growth while the latter suppressed the growth of the tumor (Fig. 5). Why are these two not correlated?
3. It is interesting to find that SLCs may mediate the uptake of A947 in hepatocytes. However, this finding was not discussed to explain why A947 accumulated at different levels in different tissues. Do tissues uptake and accumulate more A947 express higher levels of SLCs than tissues with lesser A947 uptake and accumulation?
4. Did the authors measure SMARCA2 levels in hepatocytes after A947 treatment in the presence and absence of Rifamycin SC to demonstrate that blocking A947 uptake with SLC inhibitor can also abrogate SMARCA2 degradation?

Minor concerns:

1. Abstract and page 8, lines 8-11: It might be easier to follow the data if the authors can discuss tissue distribution and accumulation of A947 in a descending manner. The reason to highlight lung for the tissue distribution is not clear considering that lung was not the tissues that showed high levels of A947 uptake and accumulation compared to many other tissues, particularly at 24h and later time points as shown in Fig. 2B.
2. Data for the Spleen is missing from Table 1.
3. Table 6 is missing from the manuscript.

COMMSMED-23-0390-T

Tissue distribution and retention drives efficacy of rapidly clearing VHL-based PROTACs

We want to thank the editors and reviewers for the excellent comments and questions that will strengthen the manuscript.

1. We have addressed each question shown below with newly added experiments and data (in associated sections in the manuscript and supplemental) for several questions.
2. Revised manuscript contained corresponding changes (in track change and clean versions).
3. Several references were added and updated in the revision.
4. We apologize for late submission of Policy Checklist.
5. All authors approved the author changes.

Responses to reviewers' questions as italic and underlined:

Referee expertise:

Referee #1: PhD, medical chemistry, PK/PD, drug discovery, cancer

Referee #2: PROTAC design, Biological Chemistry and Drug Discovery

Referee #3: PROTAC design, antitumour, oncology, pharmacodynamics

Reviewers' comments:

Reviewer #1 (Remarks to the Author):

This research manuscript described the ADME of 14C-A947 and provided a new strategy that used tissue distribution rather than traditional plasma exposure to develop a PROTAC. It is interesting that though the plasma concentration of A947 is low, the tissue concentrations and long half-life in the selective tissues are sufficient for efficacy in xenograft models. Usually, drug exposure in plasma is a very important parameter to develop, but for PROTAC drugs, the tissue concentration and half-life should be considered because of their poor physical properties due to their high molecular weights.

Response: Thank you for sharing your understanding of poor physiochemical properties of PROTACs that might lead to tissue to plasma drug asymmetry and for your positive comments.

Following major comments/suggestions noted here to authors to consider answering prior to its publication in this reputed journal:

1. In Fig.3C, the structures of 14C-A947 metabolites in the urine, bile, and feces of rats are identified. The metabolites data clearly supports the readers for further optimization to facilitate better or improved PK parameters. The liver is the main metabolic organ; authors should include

the liver as an organ to identify the metabolites in the liver or P450 and discuss the liver or P450 metabolites of A947.

Response: As the reviewer suggested, the liver is the major organ for metabolism of A947. Metabolite identification from hepatocytes and liver microsomes as well as P450 dependence were added in the revision, together with associated methods and metabolite distribution. These new additions are shown in both results section and supplemental.

2. In paragraph 2 of the “Result, Tissue distribution in rats” section”, adrenal glands, kidneys, liver, lungs, thyroid gland, pancreas, salivary glands, and spleen with a C_{max} of 17, 22, 32, 12, 18, 9, 12, and 21 µg/g, respectively (Fig. 2A, S1A, S1B, Table S1A, S1B, S1C), and the half-life ranged from 87-100 hours. Combining with Table 2 (C_{max}, AUC, half-life). So, the Liver is a key target for A947, hence authors should justify the rationale for kidney cancer, but liver cancer models seem promising for these series to consider discussing the role of A947 in liver cancers.

Response: As the reviewer suggested, the liver is also an organ for distribution of A947. In addition A947 is routed to liver for elimination. Therefore, A947 would be a great agent to target liver cancers as well as lung cancers. Text was modified to reflect on this point. The lung cancers was initially emphasized since we provided efficacy of A947 in lung cancer xenograft models. The target organ is not limited to lung. Liver and kidney are also viable organs to disease treatment by A947.

3. A947 tissue distribution kinetics via ¹⁴C-labeled Brahma-associated protein (BRM) degrader. The data shows A947 quickly distributed to tissues after IV dosing, where it is accumulating and is retained in tissues such as the lung and spleen with tissue-to-plasma ratios of >25 at 24 hours. A947 was quickly excreted in the bile of rats. These findings provide comprehensive data for cellular uptake, distribution, metabolism, and excretion of PROTAC.

The authors used IV dosing instead of oral delivery. Authors could discuss what's the difference in metabolism and pharmacokinetics between IV and Oral delivery. Authors are also suggesting optimizing PROTACs for oral drug delivery based on metabolism and pharmacokinetics data. Hence authors should state how they proceed in oral PROTACs for the A947 series.

Response: Thank you for your supporting comments. The first pass metabolism is expected if a PROTAC would be bioavailable after PO compared to IV administration. Since we did not find bioavailability of A947 after PO administration and the IV administration was an emphasized dosing strategy in the current study, we were not able to compare metabolism difference between PO and IV. We can only speculate that formulations like nano-formulations might enable PO administration (as referenced). In the revision, we pointed out that fixing metabolic softspots could be a direction for optimization. Additional revision was made on formulations.

4. For PROTAC's half-life, except for enzymatic transformation, could authors also discuss what's another potential to undergo partial non-enzymatic degradation?

Response: A947 metabolism is catalyzed by P450 enzymes and non-enzymatic degradation was minimal in in-vitro incubations. This was added in the revision.

5. Given the strong association between lipophilicity and CYP inhibition, additional scrutiny of

drug–drug interaction (DDI) potential is warranted for PROTACs in poorly precedented cLog P/Log D space. CYP inhibition was investigated for both ‘capped’ POI/E3 ligands and full PROTACs. One factor the author should be highlighted quantifying the risk of CYP inhibition.

Response: thank you for the comments. As the reviewer suggested, the DDI potential (experimental and data) was added in the revision. A947 was a direct and time-dependent inhibitor of P450 3A. It is not surprising for A947 to have the P450 inhibition potential and as a P450 substrate given the strong association between compound lipophilicity and interactions with P450 enzymes.

Reviewer #2 (Remarks to the Author):

The authors present a significant body of data profiling the absorption, distribution, metabolism and excretion of an in vivo active SMARCA2 degrader tool compound, utilising a C14 labelled analogue of a previously published SMARCA2 degrader A947. This study is the first of it's kind with respect to detailing the tissue distribution of an i.v. dosed VHL PROTAC and highlights very important learnings for the wider field that could prove to be of significant clinical benefit. Most crucially, the authors show that i.v. dosed VHL PROTACs may enable sustained efficacy in particular tissues where higher free concentration of drug is retained in comparison with plasma at long time points. There are also important learnings regarding the major routes of elimination of the molecule used here, 14C-A947. This is a very interesting and well-timed study that I would fully support for publication. It will provide an important benchmark for the field in considering when and how VHL PROTACs could be employed, particularly in oncology. A limitation of the study as presented is that it does not go on to suggest how in vitro adme discovery cascades could be reshaped to help with identifying and designing molecules that could capitalise on the features shown here, a challenge highlighted by the authors for current PROTAC optimisation approaches. Though this does not detract from the value of the data and analysis that is included.

Response: Thank you for your very positive comments. A very good suggestion to use in vitro ADME properties to help PROTAC optimization. In vitro experiments to identify metabolic softspots and drug-drug interaction potential were added in the revision, which provided some strategies for structural optimization. Additional revision was made on potential formulations.

I would in principle support publication of this manuscript as presented, though there are a few minor points that could be clarified to help the reader:

- On page 12 the following statement is made: ‘Even the estimated free drug concentrations were much higher at later time points (168 hours, K_p , could be >30). Firstly, I think the author mean $K_{p,u}$ not K_p in this context, please correct/clarify. In addition, as the authors have used free fractions of drug in various tissues in rat ex vivo studies as a basis for calculating free drug in mouse tissue, it would be helpful to include a sentence or two identifying the potential limitations of that and to what extent this limits how conclusive one can be regarding the precise free concentrations in mouse tissue. It does not detract from the overarching point that if tissue binding is constant but total drug levels are changing, then the ratio of free drug concentrations in tissue relative to plasma (i.e. $K_{p,u}$) over time must be increasing.

Response: Thank you for the important comments and recognizing importance of the free drug concentration data. Yes, It is K_p,uu and this was clarified in the revision. The limitations of using rat tissue binding to estimate the mouse free drug concentrations were pointed out in the revision. Additional protein binding data were also provided to help the estimation.

- At the beginning of the section 'Hepatocyte uptake and cancer cell retention', the authors simply say 'in the cell uptake assay...'. Whilst I appreciate some detail is given in the methods section for this assay, a brief (one or two sentence) introduction to what this assay is and how uptake of compound was detected would significantly help reader interpretation of the following section.

Response: Thank you for the comment. Introduction for the assay was added in the revision. Cell uptake mechanism of A947 was investigated by measuring cellular concentration with LC-MS/MS in hepatocytes in the presence and absence of transporter inhibitors. The compound in the cells was cleanly separated from that in the media through an oil-layer by centrifugation after incubations.

Cell retention of A947 was investigated through monitoring PD effect of SMARCA2 degradation in cell incubations after extensive washout of the compound from the incubations in the presence and absence of competing E3 ligand.

Reviewer #3 (Remarks to the Author):

The authors were the first to use ^{14}C to label PROTAC to study its tissue distribution, metabolism and clearance. Using ^{14}C labeled A947, a von Hippel-Lindau (VHL)-based Brahma associated protein (BRM) PROTAC, they found that this PROTAC rapidly distributed into tissues but accumulated and retained in high concentrations in the thyroid, spleen, kidney and liver in rats after iv administration. A947 was metabolic stable and was primarily eliminated via excretion in bile. However, elimination of tissue A947 was very slow after it accumulated in tissues. Similar findings were also observed in mice. These findings suggest that this PROTAC and other VHL-based PROTACs with similar chemical and physical properties can potentially sustain a long-term exposure to induce a PK-PD decoupling as reported previously for a VHL-based BCL-xL PROTAC (Khan S et al. Nat Med, 25: 1938-1947, 2019). These properties are highly desirable for the clinical application of these PROTACs because they allow a reduction in dosing and dosing frequency to reduce potential drug toxicity while maintaining their therapeutic efficacy. In addition, using cell culture the authors also demonstrated that SLCs may be responsible for the hepatocyte uptake of A947. Identification of a specific SLC responsible for the uptake will further advance our understanding on how PROTACs get into cells. Overall, the authors reported the first detailed study to evaluate PROTAC ADME and DMPK properties, which will produce a major impact on the future development of PROTACs and design of preclinical and clinical testing of PROTACs.

Response: Thank you for your very positive comments. The reference and your wording were used in the revision.

These findings suggest that this PROTAC and other VHL-based PROTACs with similar chemical and physical properties can potentially sustain a long-term exposure to induce a PK-PD decoupling as reported previously for a VHL-based BCL-xL PROTAC (Khan S et al. Nat

Med, 25: 1938-1947, 2019). These properties are highly desirable for the clinical application of these PROTACs because they allow a reduction in dosing and dosing frequency to reduce potential drug toxicity while maintaining their therapeutic efficacy.

Major concerns:

1. The authors reported differential tissue distribution, accumulation, and retention of A947 after iv injection in rats using QWBA and by quantification of ¹⁴C labeled A947 in excised tissues. Even though the PROTAC appears intact chemically and should be active, its activity was not tested in tissues. The authors should measure the levels of SMARCA2 in selected tissues that had high and low A947 accumulation to see if the tissue concentrations of A947 are inversely correlated with the levels of SMARCA2.

Response: Thank you for the comment. The inverse correlation between the A947 concentrations and SMARCA2 levels was demonstrated in tumor tissues. The correlation of SMARCA2 and tumor growth was further demonstrated in the xenograft models. The PD data from other tissues were not measured although an inverse relationship between drug concentration and SMARCA2 levels is anticipated.

2. In Table 4, the authors showed that 20 mg/kg A947 was equally effective as 40 mg/kg A947 in reducing the levels of SMARCA2 in tumor xenografts, but the former had no significant effect on tumor growth while the latter suppressed the growth of the tumor (Fig. 5). Why are these two not correlated?

Response: Monitoring SMARCA2 protein expression by Western blotting does not provide sufficient resolution to address maximal pathway inhibition, hence we monitored expression of a transcriptional target gene (KRT80) that is regulated by SMARCA2. We find that KRT80 is suppressed to a greater level at 40mg/kg (69% suppression) relative to the 20mg/kg (46% suppression) dose. A more detailed description of the pharmacodynamic biomarkers can be found here: <https://www.nature.com/articles/s41467-022-34562-5> (DOI: 10.1038/s41467-022-34562-5).

3. It is interesting to find that SLCs may mediate the uptake of A947 in hepatocytes. However, this finding was not discussed to explain why A947 accumulated at different levels in different tissues. Do tissues uptake and accumulate more A947 express higher levels of SLCs than tissues with lesser A947 uptake and accumulation?

Response: Thank you for the question. The impact of SLC on A947 uptake and accumulation was further discussed in the revision. In our hepatocyte uptake study, we attempted to identify SLC transporter responsible for A947 uptake by using different inhibitors specific for known drug-transporting SLCs expressed in the liver, including OCTs and OATPs. However, we have observed a significant decrease in accumulation when we use a pan-SLC inhibitor. The result suggested that A947 uptake was mediated by member(s) of the solute carriers. However, it is difficult to pinpoint which SLC(s) are responsible for the hepatic uptake and the uptake in other tissues at this point. As noted by a perspective published in Cell (<https://doi.org/10.1016/j.cell.2015.07.022>), there are over 400 SLC transporters. Many of the known drug-transporting SLCs exhibited differential tissue expression. It is reasonable to hypothesize that SLCs responsible for A947 uptake are expressed at higher levels in tissues

with more A947 accumulation. However, we can only make that statement when we identify the specific uptake mechanism for A947. We appreciate the reviewer's question, and efforts are underway to support the proposed hypothesis.

4. Did the authors measure SMARCA2 levels in hepatocytes after A947 treatment in the presence and absence of Rifamycin SC to demonstrate that blocking A947 uptake with SLC inhibitor can also abrogate SMARCA2 degradation?

Response: Thank you for the question. SMARCA2 levels were not measured as the objective of the experiment was to investigate cell uptake that was demonstrated directly by measuring its cellular concentrations by LC-MS/MS.

Minor concerns:

1. Abstract and page 8, lines 8-11: It might be easier to follow the data if the authors can discuss tissue distribution and accumulation of A947 in a descending manner. The reason to highlight lung for the tissue distribution is not clear considering that lung was not the tissues that showed high levels of A947 uptake and accumulation compared to many other tissues, particularly at 24h and later time points as shown in Fig. 2B.

Response: Thank you. The abstract and Page 8 were revised as suggested. In the revision we also add liver and kidney in addition to lung as target organ for cancer treatment. Originally the lung was emphasized since we provided A947 efficacy data in lung cancer models. The target organ is not limited to lung. Liver and kidney are also viable organs to disease treatment by A947.

2. Data for the Spleen is missing from Table 1.

Response: Thank you. It is added.

3. Table 6 is missing from the manuscript.

Response: Thank you. It was Table 5 and It is corrected in the revision.

REVIEWERS' COMMENTS:

Reviewer #1 (Remarks to the Author):

To Authors:

The authors responded with rationale and some additional dates presented in the revised manuscript. This reviewer recommended it to the editorial board for publication.

Reviewer #2 (Remarks to the Author):

Many thanks for addressing my requests and congratulations on a great piece of work. I would fully support publication of this manuscript.

Reviewer #3 (Remarks to the Author):

The manuscript has been nicely revised. All my concerns have been adequately addressed by the revision.

REVIEWERS' COMMENTS:

Reviewer #1 (Remarks to the Author):

To Authors:

The authors responded with rationale and some additional data presented in the revised manuscript. This reviewer recommended it to the editorial board for publication.

Reviewer #2 (Remarks to the Author):

Many thanks for addressing my requests and congratulations on a great piece of work. I would fully support publication of this manuscript.

Reviewer #3 (Remarks to the Author):

The manuscript has been nicely revised. All my concerns have been adequately addressed by the revision.

Responses:

We want to thank the editors and reviewers for your excellent comments.

We have provided point-to-point response to all inquiries in CommMed_AIP_Tables.

The following main changes were made in this new revision.

1. Multiple experimental sections were moved from Supplementary Information to the main text
2. Fig. 7 that was the visual abstract from previous version is incorporated in discussion.
3. Abstract was modified into 4 sections background, Methods, Results, and Discussion plus a plain language summary
4. The individual numerical data to all line- and bar-graphs are provided in Supplementary Information
5. 10 Figs plus tables are referenced in the text

These responses were also included in the CommMed_AIP_Tables.